# The Truth Stays in the Family:
# Enhancing Contextual Truthfulness via Inherited Heads in Model Lineages

Miso Choi [1]  Seonga Choi [1]  Mincheol Kwon [1]  Woosung Joung [1]  Jinkyu Kim [1 2]  Jungbeom Lee [1]

## Abstract

Recent advances in large language models (LLMs) have produced many specialized multimodal LLMs (MLLMs) that share common foundational LLMs, forming distinct model lineages. It remains unclear whether a fundamental behavioral link exists between the foundational LLMs and downstream variants. We investigate this question by quantifying head-level context-truthfulness scores. Across diverse LLM and MLLM lineages, including Vicuna-, Qwen2.5-, LLaMA2-, and Mistral-based models, we find that Truth Scores are strongly preserved within model families, even after instruction tuning or multimodal adaptation. We further show that this inheritance is consistent with attention-head weight preservation, and that context-truthful heads attend to query-relevant evidence. Building on this finding, we propose TruthProbe, a soft-gating strategy that amplifies context-truthful heads while preserving other head contributions. TruthProbe improves contextual truthfulness on HaluEval and reduces multimodal hallucination on POPE and CHAIR, with base-LLM Truth Scores transferring effectively to their fine-tuned LLM and MLLM descendants. Code is available at https://github.com/miso-choi/TruthProbe.

## 1. Introduction

Recent advancements in large language models (LLMs) (Minaee et al., 2024) have given rise to a wide range of specialized models (Yin et al., 2024; Caffagni et al., 2024), all of which originate from a core foundational LLMs. This pattern reflects a broader trend: rather than building entirely new models from scratch, base LLMs are often refined through fine-tuning or multimodal extensions to serve domain-specific needs—ranging from mathematical reasoning (Yang et al., 2024) to vision-language understanding (Zhang et al., 2025), or even multi-sensory processing (Xu et al., 2025). Such evolutionary trajectories highlight that many advanced multimodal LLMs (MLLMs) share a clear lineage with their base LLMs.

While these architectural advancements have significantly expanded the functional capabilities of LLMs and MLLMs, their deployment to the real world remains challenged by a critical bottleneck: hallucination (Kalai et al., 2025; Zhou et al., 2024)—the generation of factually incorrect or contextually inconsistent information. As these models move from research benchmarks to real-world integration, such as autonomous driving and user's decision making process, ensuring their reliability has become a paramount concern. This raises a fundamental, yet unexplored, question:

*Does a fundamental behavioral link exist between derived MLLMs and their foundational LLMs?* Uncovering this connection could enable the development of a systemic framework for enhancing truthfulness across an entire model family. While prior work has introduced various hallucination-mitigation strategies (Zhou et al., 2024; Yang et al., 2025b; Huang et al., 2024; Xing et al., 2024; Li et al., 2023c; Lyu et al., 2024), they often treat models as isolated instances, overlooking the persistent functional roles that specific attention heads may maintain when a base LLM is adapted into downstream variants. We hypothesize that specific attention heads are specialized in encoding context-faithful information and that this "truthfulness trait" is preserved within model lineages. To test this, we employ a linear probing methodology, inspired by previous work (Li et al., 2023c), to quantify the degree of context-truthfulness across different attention heads. We primarily analyze the Vicuna-7B (Chiang et al., 2023) and Qwen2.5 (Qwen et al., 2025) families, including their multimodal descendants such as LLaVA-1.5 (Liu et al., 2024a), LLaVA-NeXT (Liu et al., 2024b), Qwen2.5-VL-Instruct (Bai et al., 2025b), and Qwen2.5-VL-Omni (Xu et al., 2025). Beyond these representative MLLM families, we further examine fine-tuned LLM lineages, such as Qwen2.5-7B/Qwen2.5-7B-Instruct

---

[1]Korea University, Seoul, Republic of Korea [2]Kakao Mobility Corp., Seongnam, Republic of Korea. Correspondence to: Jungbeom Lee <jbeomlee@korea.ac.kr>.

*Proceedings of the $43^{rd}$ International Conference on Machine Learning*, Seoul, South Korea. PMLR 306, 2026. Copyright 2026 by the author(s).

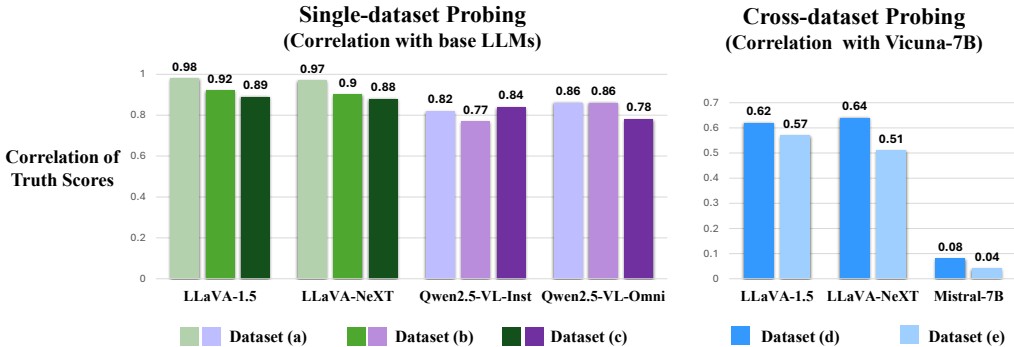

*Figure 1.* **Correlation of Truth Scores under Single- and Cross-dataset Probing.** *Left:* Within the same model family, Truth Scores remain highly correlated between base LLMs and their multimodal descendants across different probing setups. *Right:* This alignment persists under cross-dataset probing within the Vicuna family, whereas the unrelated Mistral-7B shows near-zero correlation. For probing dataset setups, please refer to Tab. 5 and Tab. 6 in the Appendix A.

and LLaMA2/Vicuna-7B, as well as additional multimodal lineages including Mistral-7B-Instruct-v0.2/LLaVA-Med in the Appendix. This broader evaluation allows us to examine whether head-level context-truthfulness is preserved not only in a single model family, but across diverse fine-tuned LLM and MLLM lineages.

Our analysis reveals the key property within model families: **Inheritance.** We find that MLLMs exhibit a high correlation in Truth Scores with their base LLMs, despite additional multi-modal training. Remarkably, even when models are probed using entirely different data sources, those belonging to the same model family maintain substantially higher truthfulness correlations compared to unrelated families. This finding suggests that the truthfulness-related behavior of attention heads is largely preserved when a base LLM is adapted into downstream variants. We further provide mechanistic evidence for this inheritance by analyzing parameter-level weight drift: within-family fine-tuning induces only minor changes to attention-head weights, whereas unrelated model families exhibit substantially larger parameter-space divergence. This weight-preservation pattern offers a plausible explanation for why Truth Score distributions remain aligned within a lineage but collapse across unrelated families.

Building on these insights, we propose **TruthProbe**, a soft-gating strategy that leverages the obtained Truth Scores to amplify the influence of context-truthful heads, thereby ensuring that the model's final outputs are more faithfully grounded in the given context. Unlike hard masking, our approach preserves the contribution of all heads while softly increasing the influence of heads with stronger context-truthfulness signals. Rather than simply evaluating this mechanism on a single model, we demonstrate that it generalizes consistently across models sharing the same backbone. In particular, Truth Scores derived from a base LLM can act as a "plug-and-play" soft gate for downstream models

in the same lineage, including instruction-tuned LLMs and MLLMs.

Our results across HaluEval (Li et al., 2023b), POPE (Li et al., 2023d), and CHAIR (Rohrbach et al., 2018) benchmarks suggest that we have identified a shared reliability mechanism: a lightweight plug-in gating mechanism that remains effective across multimodal and fine-tuned variants derived from the foundational model. On LLMs, Truth-Probe improves contextual truthfulness on HaluEval; on MLLMs, it improves object-presence reasoning on POPE and reduces object hallucination in image captioning on CHAIR. We further show that Truth Scores obtained from a base LLM achieve comparable gains to those obtained by directly probing the corresponding MLLM, demonstrating their transferability across model lineages. In addition, our attention-overlay analysis shows that context-truthful heads attend to query-relevant visual evidence, whereas low-truthfulness heads tend to exhibit weakly semantic attention patterns. Together, these results indicate that inherited Truth Scores capture functionally meaningful grounding behavior, rather than serving merely as statistical artifacts.

By uncovering this inheritance, we provide a principled foundation for improving truthfulness across entire model families, moving beyond individual fixes toward a more systemic understanding of model reliability.

Our contributions are summarized as follows:

- **Identifying the Identity of Context-Truthful Heads.** We measure how well each transformer head grounds responses in the context, yielding a Context-Truthfulness Score (Truth Score).

- **Discovering the Inheritance of Context-Truthful Heads.** Single- and cross-dataset analyses show that Truth Scores are strongly correlated within model families, indicating preservation of context-truthful heads

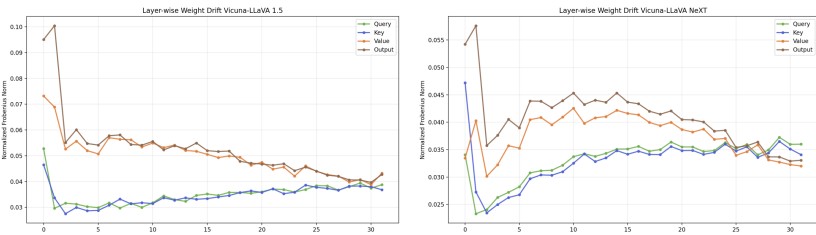

Figure 2. **Layer-wise weight drift** measured by the Frobenius norm of weight differences. (Left) Vicuna-7B vs. LLaVA-1.5, (Right) Vicuna-7B vs. LLaVA-NeXT.

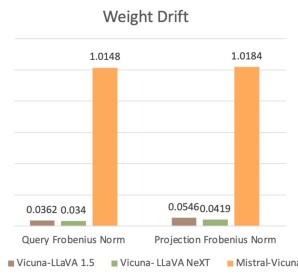

Figure 3. **Comparison of weight drift** between within-family models (Vicuna-7B vs. LLaVA-1.5/LLaVA-NeXT) and cross-family models (Vicuna-7B vs. Mistral-7B).

when base LLMs are fine-tuned into LLMs or MLLMs.

- **Providing Mechanistic Evidence for Inheritance and Behavioral Analysis of Truthful Heads.** We show that this inheritance is consistent with parameter-level weight preservation within model families, and further verify through attention-overlay analysis that context-truthful heads attend to query-relevant evidence.

- **Soft-Gating for Truthfulness Enhancement.** We propose TruthProbe, a soft-gating strategy using Truth Scores to improve model truthfulness, and demonstrate that Truth Scores from base LLMs can be effectively transferred to fine-tuned LLMs and MLLMs, yielding gains on HaluEval, POPE, and CHAIR.

## 2. Discovering Inherited Truthful Heads

### 2.1. Measuring Head-level Context Truthfulness

We aim to identify attention heads that support context-grounded truthful reasoning. While prior work (Li et al., 2023c; Baek et al., 2026) shows that truthfulness-related concepts can be represented in activation space, our focus is different: *we ask whether individual attention heads faithfully incorporate the provided context, rather than merely retrieving parametric knowledge*. This distinction is particularly important for MLLMs, where a truthful answer often depends on grounding in the given visual evidence.

To quantify this property, we define a head-level Truth Score. For this setting, we structure the input as $x = \{x_{context}, x_{question}, x_{answer}\}$, where $x_{context}$ can be text of world knowledge or the real image and $x_{answer}$ can be truthful answers or hallucinated ones. We probe the activations at the final answer token, based on the assumption that, in an auto-regressive model, this position encodes the accumulated features from all preceding tokens. The probe of each head is trained as a binary classifier to determine whether the head reliably incorporates the given context or

contributes misleading information. The validation accuracy of this probe is used as the *Truth Score* of the head.

A high Truth Score indicates that the head output at the final answer token contains linearly decodable signals that distinguish truthful answers from hallucinated ones, given the preceding context. Thus, the Truth Score serves as a diagnostic measure of how strongly each head representation reflects truthfulness-relevant information conditioned on the given context. We use these scores in two ways: first, to analyze whether such truthfulness-relevant head representations are preserved within model families, and second, to construct the soft head-gating mechanism to enhance models' truthfulness. The full probing protocol, including dataset construction, train–validation split, and cross-validation, is provided in Appendix A.

### 2.2. Truthful Heads Are Preserved Within Model Families

We next examine whether the head-level truthfulness structure identified in a base LLM is preserved after the model is adapted into multimodal descendants. Specifically, we ask whether the heads that contain truthfulness-relevant signals in a foundational LLM remain aligned with those in its fine-tuned MLLMs.

To answer this question, we analyze two representative model families: the Vicuna family, including Vicuna-7B (Chiang et al., 2023), LLaVA-1.5 (Liu et al., 2024a), and LLaVA-NeXT (Liu et al., 2024b), and the Qwen2.5 family, including Qwen2.5-7B (Qwen et al., 2025), Qwen2.5-VL-Instruct (Bai et al., 2025b), and Qwen2.5-VL-Omni (Xu et al., 2025). For each model, we compute head-level Truth Scores and compare their distributions across models within the same family.

As shown in Fig. 1, Truth Scores exhibit strong within-family alignment. In the single-dataset setting, base LLMs and their multimodal descendants show consistently high correlations, ranging from approximately 0.77 to 0.98. This

**(1) Inheritance of Context-Truthful Heads**

**(2) Soft-Gating for Truthfulness Enhancement**

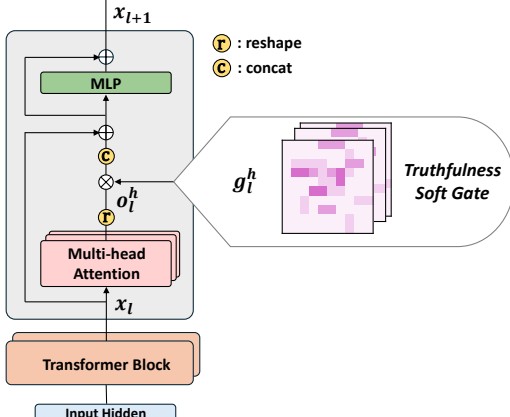

*Figure 4.* **(1) Heatmaps of head-level Truth Scores for two model families.** Vicuna-based models (Top) and Qwen2.5-based models (Bottom). The heatmaps visually confirm that finetuned MLLMs exhibit Truth Score distributions highly consistent with those of their foundational LLMs, suggesting an **inheritance** of truthfulness traits. **(2) Overview of the proposed TruthProbe mechanism.** Soft gating refines the residual pathway by modulating individual head contributions based on their estimated Truth Scores, thereby promoting context-truthful reasoning.

indicates that multimodal fine-tuning largely preserves the head-level structure associated with context-truthfulness. The preservation is observed not only when LLMs and MLLMs are probed with comparable textual inputs, but also when the probing setup includes multimodal inputs.

We further evaluate this inheritance under a cross-dataset setting, where LLMs and MLLMs are probed using different datasets and modalities. Although this setting introduces a stronger distribution shift, within-family correlations remain substantially higher than cross-family correlations. For example, Vicuna-7B and its LLaVA descendants maintain correlations of approximately 0.51–0.64, whereas Mistral-7B (Jiang et al., 2023), an unrelated model family, shows near-zero correlation (0.04–0.08) with Vicuna-7B. This contrast suggests that truthful heads are not universally shared across independently pretrained models, but are instead organized in a lineage-specific manner.

Taken together, these results reveal an inheritance property of context-truthful heads: *fine-tuned MLLMs preserve the head-level truthfulness structure of their foundational LLMs.* This finding motivates a family-level intervention strategy, where Truth Scores estimated from a base LLM can be reused to guide its downstream LLM or MLLM variants without probing each descendant model from scratch. Detailed probing datasets, correlation computation, and dataset-specific settings are provided in Appendix A.

### 2.3. Mechanistic Evidence: Inheritance Follows Weight Preservation

To better understand why truthful heads are preserved, we analyze the layer-wise weight differences between the base LLM, Vicuna-7B, and its multimodal variant, LLaVA-1.5 and LLaVA-NeXT. We measure layer-wise Frobenius norm of weight differences, as in Fig. 2, showing drift is concentrated in early layers.

Prior work (Zheng et al., 2024) suggests early layers process input signals, whereas deeper layers handle reasoning and high-level semantic integration. In our analysis, we find that truthful heads—identified by high Truth Scores—are predominantly located in middle to deeper layers (e.g., 80.0% of Top-20 truthful heads in LLaVA-1.5 are located in layers 10-31.), indicating that they are associated with context-level reasoning rather than low-level feature extraction. Combining these, we argue that truthful heads are preserved as they reside in minimally modified layers, leading to the observed inherited Truth Scores.

We further quantify the overall degree of parameter drift using the Frobenius norm of weight differences averaged across layers and heads, as shown in Fig. 3. We find a clear contrast: within the same family (e.g., Vicuna-7B $\rightarrow$ LLaVA-1.5 / LLaVA-NeXT), the averaged Frobenius norm is extremely small ($\approx 0.03$), indicating that the attention-head parameters are largely preserved during fine-tuning. In contrast, across unrelated families (e.g., Vicuna-7B vs. Mistral-7B), the norm is substantially larger ($\approx 1.01$), reflecting a much larger architectural/parameter-space divergence. This parameter-level preservation provides a plau-

sible explanation for why head-level Truth Score patterns remain aligned within a model family, while such alignment collapses across unrelated families.

This observation is also consistent with prior findings (Zaken et al., 2022; Hu et al., 2022; Aghajanyan et al., 2021) that Transformer fine-tuning tends to induce low-rank and localized updates, modifying only a small subset of parameters while preserving much of the original structure.

This result supports our interpretation at the parameter level: (1) within-family fine-tuning induces only minor changes to the attention-head weights, providing a plausible explanation for why Truth Score patterns remain aligned within a lineage. (2) In contrast, unrelated model families exhibit substantial parameter-space divergence, which is consistent with the near-zero cross-family correlation of Truth Scores observed in Sec. 2.2.

## 3. TruthProbe: leveraging Inherited Truth Scores for Soft Head Gating

Building on the analyses in Sec. 2.1 and 2.2, we introduce **TruthProbe** as illustrated in Fig. 4, a refinement strategy that uses the identified Truth Scores of attention heads to guide model behavior. TruthProbe selectively increases the influence of highly truthful heads and attenuates less reliable ones, steering the residual stream toward context-faithful signals. This targeted adjustment aims to improve the overall truthfulness of models without altering their core architecture.

**Soft Head Gating for Truthfulness Amplification.**
To further refine the residual pathway with respect to context-faithful reasoning, we propose a soft gating mechanism that amplifies or attenuates the contribution of each attention head according to its estimated truthfulness score. Unlike hard masking, which discards information from untrusted heads, our approach preserves the expressive capacity of multi-head attention (MHA) while softly steering the residual stream toward reliable signals.

Formally, in a Transformer layer $l$, the attention output $o_l^h \in \mathbb{R}^d$ is modulated before the residual connection, as presented in Fig. 4 (2). To apply the Truth Score as a soft gate, $o_l$ is reshaped into head-wise components $o_l^h \in \mathbb{R}^{n_h \times d_h}$, where $n_h$ and $d_h$ denote the number of heads and the head dimension, respectively. Each component is then scaled by its corresponding gate value $g_l^h \in \mathbb{R}^{n_h}$. The gated representations are subsequently concatenated and added back to the residual stream, thereby modulating each head's contribu-

tion according to its Truth Score:

$$x_{l+1} = x_l + \text{Concat}_{h=1}^{H}(g_l^h \cdot o_l^h), \quad (1)$$

$$g_l^h = 1 + \lambda \cdot \text{norm}(S), \quad (2)$$

Here, $g_l^h$ denotes the soft gate for head $h$ at layer $l$, parameterized by the normalized Truth Score $S$ and scaled by a parameter $\lambda$. Specifically, when the norm-based score $S$ is larger, the corresponding head output is amplified beyond the baseline level, whereas smaller values reduce its relative impact. This formulation enables the model to selectively strengthen more reliable heads while suppressing less informative ones. Importantly, the proposed soft gating mechanism ensures that all heads remain active; their influence on the residual connection is adaptively modulated in proportion to their truthfulness score, thereby preserving diversity while promoting context-faithful reasoning.

By embedding this gating mechanism into the residual update, the model effectively prioritizes trustworthy contextual cues without sacrificing the diversity of representations contributed by different heads. This design allows Multimodal Large Language Models (MLLMs) to more faithfully propagate context-grounded information and mitigates the propagation of misleading or hallucinated activations.

## 4. Experiments

In this section, we focus on the core evaluation of TruthProbe: validating Truth Scores on LLMs, transferring base-LLM Truth Scores to finetuned MLLMs, and extending the same transfer to finetuned LLMs.

Beyond these main results, we provide extensive additional experiments and analyses in the Appendix. We include comparison with ITI (Li et al., 2023c) (§ I), the ablation of attention-head gating (§ F), the practical benefit of our approach (§ G), further attention-pattern visualizations of truthful heads (§ H). Finally, we investigate statistical robustness (§ J), and low-resource generalization (§ L), cross-family alignment (§ K) and extend the evaluation to additional model families and sizes (§ M), and benchmarks (§ N).

### 4.1. Experimental Setting

**Baseline Models.** To investigate the transferability of truthfulness heads across model families, we focus on models that share a common backbone. Specifically, we use Vicuna-7B (Chiang et al., 2023) as the base LLM and evaluate its fine-tuned counterparts, LLaVA-1.5 (Liu et al., 2024a) and LLaVA-NeXT (Li et al., 2024). In parallel, we conduct experiments on the Qwen2.5 family, comparing the base Qwen2.5 (Qwen et al., 2025) model with its vision–language variants, Qwen2.5-VL-Instruct (Bai et al., 2025b) and Qwen2.5-VL-Omni (Xu et al., 2025). For experiments on the inheritance of truthfulness in fine-tuned

LLMs, we also include instruction-tuned models: Qwen2.5-7B-Instruct and Vicuna-7B, whose respective base LLMs are Qwen2.5-7B and LLaMA2-7B (Touvron et al., 2023). This setup allows us to systematically analyze whether the identified truthful components remain consistent when models are adapted to multimodal tasks or instruction-finetuned LLMs within the same architectural lineage.

**Probing Dataset for Truth Scores used in Soft Gating.** For Truth Scores used in Soft Gating, we use two probing datasets: a subset (292 samples) of HaluEval (Li et al., 2023b) for LLM Truth Scores; and RLHF-V (Yu et al., 2024), using only its question–answer split (2,726 samples), for MLLM Truth Scores. We use a larger dataset for MLLMs because their visual processing produces substantially more tokens, requiring more samples to obtain stable and reliable Truth Scores. All Truth Scores are computed using 5-fold cross-validation to ensure robustness.

**Evaluation Benchmarks for Hallucination Mitigation.** HALUEVAL (Li et al., 2023b) is a large-scale hallucination benchmark composed of task-specific datasets (e.g., QA) generated from sources such as HotpotQA (Yang et al., 2018), and general user queries paired with multiple LLM responses. We use the question-answering split, where the model must distinguish factual answers from hallucinated ones. For our setting, 292 samples are used for linear probing to obtain Truth Scores, and evaluation for Tab. 1, 4 is performed on the remaining 9,708 samples. Since answer selection is randomized in the original pipeline, we construct three evaluation sets using different random seeds and report the mean across them.

**POPE** (Li et al., 2023d) is designed to assess whether MLLMs accurately identify object presence in images through a binary classification format. We evaluate our method on POPE, which leverages data sourced from MSCOCO (Lin et al., 2014) and A-OKVQA (Marino et al., 2019). For each dataset source, we report the mean of three splits: *random*, *popular*, and *adversarial*.

**CHAIR** (Rohrbach et al., 2018) is designed to evaluate object hallucination in image captioning task. It comprises of two standard metrics: $CHAIR_I$, the proportion of object mentions that are hallucinated, and $CHAIR_S$, the proportion of sentences that contain hallucinated objects. We randomly sampled 500 images from COCO 2014 validation set.

**Implementation Details.** All model outputs are generated using greedy decoding. For the soft gating mechanism, we use scaling parameter $\lambda$ and a normalization method to control the effect of the Truth Score. Specifically, we use centered normalization for HaluEval and CHAIR benchmarks, and min-max normalization for POPE. We adopt identical $\lambda$ values across the different POPE data sources to

*Table 1.* **Validation of Truth Scores.** Comparison between vanilla LLM models and our truth-enhanced models on the HALUEVAL benchmark, where Truth Scores are obtained via Linear Probing.

| | **HaluEval** | | | |
|---|---|---|---|---|
| **Model** | **Acc** | **F1** | **Prec** | **Rec** |
| Vicuna-7B (Chiang et al., 2023) | | | | |
| Baseline | **38.89** | 13.37 | 22.93 | 9.44 |
| + **TruthProbe**_LLM_ | 38.53 | **29.15** | **34.38** | **25.30** |
| Qwen2.5 (Qwen et al., 2025) | | | | |
| Baseline | 27.65 | 36.69 | 32.60 | 41.96 |
| + **TruthProbe**_LLM_ | **35.04** | **46.54** | **39.52** | **56.59** |

*Table 2.* **TruthProbe performance in finetuned MLLMs on POPE.** TruthProbe_LLM_ uses Truth Scores obtained from each model's base LLM (Vicuna-7B for LLaVA-1.5 and LLaVA-NeXT; Qwen2.5 for Qwen2.5-VL-Inst and Qwen2.5-VL-Omni). TruthProbe_MLLM_ uses Truth Scores derived directly from the corresponding MLLMs. (Bold = best, Underline = second best.)

| | **POPE(COCO)** | | | **POPE(A-OKVQA)** | | |
|---|---|---|---|---|---|---|
| Method | Acc | F1 | Rec | Acc | F1 | Rec |
| **LLaVA-1.5** (Liu et al., 2024a) | | | | | | |
| Baseline | **86.9** | **85.8** | 79.1 | **86.3** | **86.5** | 87.8 |
| + **TruthProbe**_LLM_ | 86.7 | **85.8** | **80.1** | 85.7 | _86.3_ | **90.1** |
| + **TruthProbe**_MLLM_ | _86.8_ | **85.8** | _79.6_ | _86.1_ | **86.5** | _89.0_ |
| **LLaVA-NeXT** (Li et al., 2024) | | | | | | |
| Baseline | 87.7 | 86.5 | 78.8 | _87.4_ | 87.4 | 86.8 |
| + **TruthProbe**_LLM_ | **88.3** | **87.3** | **80.9** | **87.7** | **88.0** | **89.7** |
| + **TruthProbe**_MLLM_ | _88.2_ | _87.2_ | _80.1_ | **87.7** | _87.9_ | _89.5_ |
| **Qwen2.5-VL-Instruct** (Bai et al., 2025b) | | | | | | |
| Baseline | 87.6 | 86.3 | 78.2 | 87.4 | 87.2 | 86.0 |
| + **TruthProbe**_LLM_ | **88.1** | **87.0** | _79.9_ | **87.8** | **87.8** | **87.7** |
| + **TruthProbe**_MLLM_ | **88.1** | **87.0** | **80.0** | _87.7_ | _87.7_ | _87.4_ |
| **Qwen2.5-VL-Omni** (Xu et al., 2025) | | | | | | |
| Baseline | 85.1 | 84.7 | 75.0 | 87.0 | 87.4 | 84.7 |
| + **TruthProbe**_LLM_ | **87.3** | **86.0** | **77.7** | **87.8** | **87.8** | **87.1** |
| + **TruthProbe**_MLLM_ | _87.1_ | _85.7_ | _77.3_ | _87.7_ | _87.6_ | _86.7_ |

ensure reproducibility. Detailed settings are provided in the Appendix C.

## 4.2. Evaluation of the Proposed Methods

**Validation of Truth Scores.** To validate the effectiveness of our proposed TruthProbe, we first validate their impact of enhancing truthfulness on LLMs. We obtain the Truth Scores for each LLMs—Vicuna-7B and Qwen2.5—by performing linear probing on a subset of the HaluEval dataset as in Section 2.1. These scores are then applied as a soft gate to the same model. We evaluate the models' truthfulness on the remaining portion of the HaluEval benchmark, ensuring a clean evaluation without any leakage from the probing phase. As demonstrated in Table 1, applying our method

significantly enhances performance, with the models showing an improved ability to judge the truthfulness of given sequences. These results highlight two takeaways: (i) the increased performance by applying a model's own Truth Scores back to itself validates that the scores accurately capture truthfulness, and (ii) even a small probing subset is sufficient to identify and reweight head-level signals to better ground the model in the given context.

**Refining Finetuned MLLMs using Truth Scores.** Building upon our findings that the Truth scores of base LLMs and their finetuned MLLMs are highly correlated—even finetuned or probed with different modalities—we explored the transferability of Truth Scores within model families. We applied the Truth Scores obtained from the base LLMs (Vicuna-7B and Qwen2.5) as a soft gate to their corresponding finetuned MLLMs. Our experiments included LLaVA-1.5 and LLaVA-NeXT (finetuned from Vicuna-7B), as well as Qwen2.5-VL-Instruct and Qwen2.5-VL-Omni (finetuned from Qwen2.5).

In Tab. 2, we evaluated TruthProbe on the POPE benchmark and observe improved performance over the vanilla models in most cases. Performance gains are primarily reflected in the Recall metric, demonstrating that our soft gate amplifies the contributions of context-faithful heads while maintaining the influence of the remaining heads.

Furthermore, we assess the effectiveness of our method in generating context-faithful image descriptions on the CHAIR benchmark (Tab. 3). The reduced hallucination rates (lower values indicate fewer hallucinations) demonstrate that our approach enhances truthfulness not only in multimodal QA, but also in text generation tasks.

In both results (Tab. 2, 3), the performance of TruthProbe$_{MLLM}$ was comparable to that of TruthProbe$_{LLM}$. This result suggests that Truth Scores obtained from base LLMs can be effectively transferred to their finetuned MLLM counterparts. It also highlights the potential for a unified approach: leveraging the Truth Scores from a single base LLM to enhance the truthfulness of multiple specialized MLLMs derived from the same foundation.

**Refining Finetuned LLMs using Truth Scores.** We use instruction-finetuned LLMs—Qwen2.5-7B-Instruct and Vicuna-7B—as baselines, with Qwen2.5-7B and LLaMA2-7B as their respective base LLMs. Truth Scores are obtained by probing each base LLMs on a subset and applied to the finetuned models, with evaluation conducted on the remaining portion of the HaluEval benchmark, using the same experimental setup as in Tab. 1. Results in Tab. 4 indicate that applying the TruthProbe from the base LLM significantly improves the model's ability to discern contextual truthfulness. Notably, TruthProbe$_{Base LLM}$ to Vicuna-7B

*Table 3.* **TruthProbe performance in finetuned MLLMs on CHAIR.** Results on object hallucination in image description setting, where models are prompted with "Please describe this image in detail." (max 64 tokens). Performance is measured using CHAIR$_I$ and CHAIR$_S$, where lower values indicate fewer hallucinated objects. (Bold = best, Underline = second-best.)

| | CHAIR | |
|---|---|---|
| Method | CHAIR$_I$ ($\downarrow$) | CHAIR$_S$ ($\downarrow$) |
| **LLaVA-1.5** (Liu et al., 2024a) | | |
| Baseline | 6.99 | 23.00 |
| + TruthProbe$_{LLM}$ | **5.36** | **17.40** |
| + TruthProbe$_{MLLM}$ | 6.20 | 21.60 |
| **LLaVA-NeXT** (Li et al., 2024) | | |
| Baseline | 6.91 | 13.40 |
| + TruthProbe$_{LLM}$ | **4.94** | **11.20** |
| + TruthProbe$_{MLLM}$ | 6.56 | 12.60 |
| **Qwen2.5-VL-Instruct** (Bai et al., 2025b) | | |
| Baseline | 6.14 | 13.20 |
| + TruthProbe$_{LLM}$ | 5.56 | 12.20 |
| + TruthProbe$_{MLLM}$ | **5.26** | **7.80** |
| **Qwen2.5-VL-Omni** (Xu et al., 2025) | | |
| Baseline | **5.26** | 11.40 |
| + TruthProbe$_{LLM}$ | 5.94 | **10.80** |
| + TruthProbe$_{MLLM}$ | 5.54 | 11.00 |

*Table 4.* **TruthProbe performance in finetuned LLMs on HaluEval.** We compare vanilla instruction-tuned LLMs with their truth-enhanced models (TruthProbe$_{Base LLM}$), where the Truth Scores are derived from the corresponding base LLMs—Qwen2.5 for Qwen2.5-7B-Instruct, and LLaMA2-7B for Vicuna-7B.

| | HaluEval | | | |
|---|---|---|---|---|
| Method | Acc | F1 | Prec | Rec |
| **Qwen2.5-7B-Instruct** (Bai et al., 2025b) | | | | |
| Baseline | 34.90 | 16.29 | 22.79 | 12.68 |
| + TruthProbe$_{Base LLM}$ | **37.35** | **17.24** | **25.36** | **13.05** |
| **Vicuna-7B** (Chiang et al., 2023) | | | | |
| Baseline | 38.89 | 13.37 | 22.93 | 9.44 |
| + TruthProbe$_{Base LLM}$ | **48.47** | **57.17** | **48.90** | **68.82** |

significantly improves performance, even surpassing the results obtained by applying Truth Scores derived from the finetuned Vicuna-7B itself (refer Tab. 1). This indicates that truthfulness inheritance emerges not only in fine-tuned MLLMs, but also in fine-tuned LLMs.

## 5. Analysis

**Truthful Heads Attend to Query-Relevant Evidence.** To investigate this, we analyze where different heads attend in the image by visualizing their attention patterns. A direct visualization of the original attention maps, however,

*Figure 5.* **Attention pattern comparison between truthful and non-truthful heads.** The left three columns show the Top-3 truthful heads, while the right three columns show the Bottom-3 (non-truthful) heads. The first row presents the 24×24 attention maps (from the final query to visual tokens), and the second row shows the corresponding attention overlaid on the image.

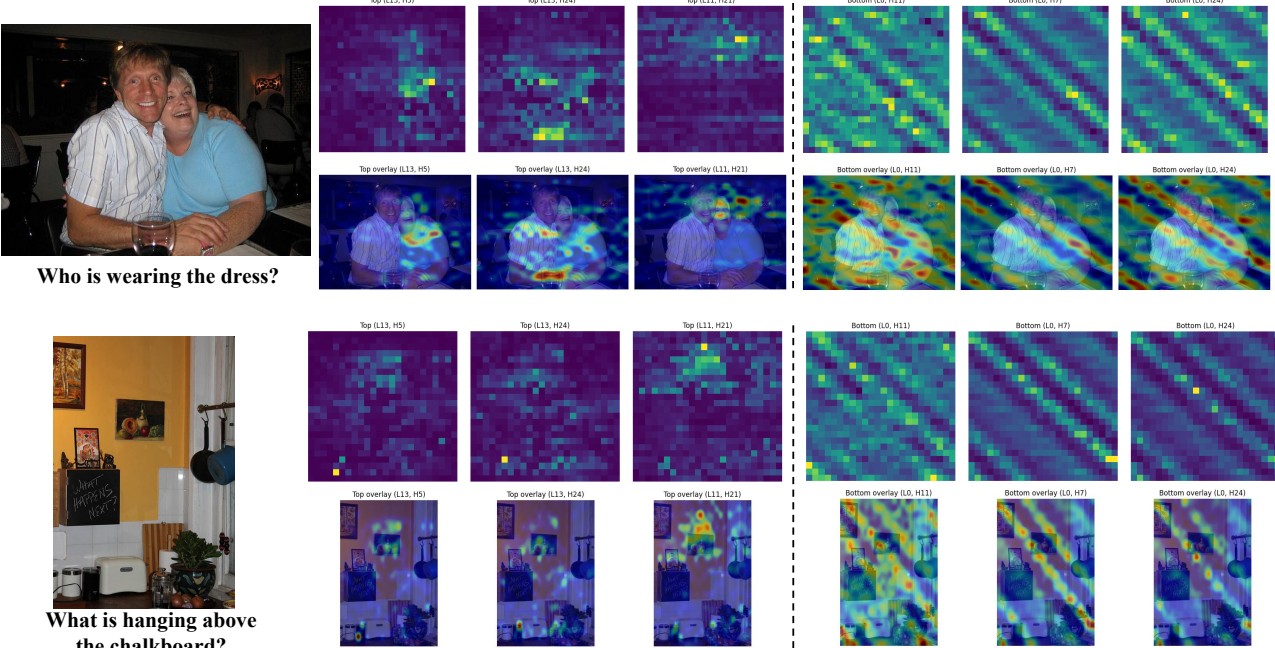

is dominated by attention sink tokens, i.e., tokens that consistently receive high attention regardless of the query. This phenomenon has been discussed in prior work (Darcet et al., 2024), which shows that Transformers tend to aggregate attention into a small set of irrelevant tokens. As a result, raw attention maps obscure head-specific behaviors.

To address this issue, we adopt the relative attention formulation of (Khayatkhoei et al., 2025). Specifically, we normalize the attention induced by a given query (e.g., "Who is wearing the dress?") with respect to a general query (e.g., "Write a general description of the image."), allowing us to isolate query-dependent attention patterns.

In Fig. 5, we visualize both attention maps (obtained from LLaVA-1.5) and their overlays on the image. Using the Truth Score (obtained by probing the base LLM, Vicuna-7B), we compare Top-k (k=3) truthful heads and Bottom-k (k=3) non-truthful heads in LLaVA-1.5. The visualization corresponds to attention from the final query token to visual tokens (24×24 grid).

From these results, we identify two distinct attention behaviors that separate truthful heads from non-truthful heads.

**1. Truthful heads exhibit semantically meaningful and query-dependent attention.** They focus on regions directly relevant to the query (e.g., the referenced object or person) and show spatially selective patterns that depend on the query, aligning with the importance of region-level understanding in vision-language models (Lee et al., 2024). This indicates their role in grounding the model's responses by attending to credible visual evidence, rather than amplifying generic features.

**2. Non-truthful heads exhibit largely position-dependent attention patterns that are weakly tied to the query semantics.** Their attention often forms diagonal or striped structures that remain nearly unchanged across different queries. Since these heads are mostly located in early layers (e.g., layer 0), where cross-modal alignment is limited, their behavior appears to reflect positional or structural biases in the visual grid rather than query-specific visual grounding.

Overall, this comparison highlights a clear mechanistic distinction: truthful heads demonstrate query-dependent, evidence-focused attention, whereas non-truthful heads largely reflect early-stage representation processing rather than direct semantic grounding.

## 6. Related Works

### 6.1. Hallucination Mitigation in Large and Vision Language Models

Hallucination in Multi-modal Large Language Models (MLLMs) refers to the generation of text that is inconsistent

with the visual input, and numerous studies (Zhou et al., 2024; Yang et al., 2025b; Huang et al., 2024; Park et al., 2025; Xing et al., 2024) have analyzed its causes and proposed methods to address it. Recent literature identifies several underlying causes for these errors, including the inherent limitations of training and evaluation procedures rewarding guessing over acknowledging uncertainty (Kalai et al., 2025) and over-reliance on statistical subsequence associations rather than faithful ones (Sun et al., 2025). These initial errors often "snowball" as the model attempts to maintain self-consistency in auto-regressive generation (Zhang et al., 2024b). To mitigate these problems, one prominent line of research focuses on training-free interventions during the inference stage. These methods (An et al., 2025; Huo et al., 2025; Wang et al., 2025; Duan et al., 2025) manipulate the output distribution or apply self-correction mechanisms without modifying the model's weights. Alternatively, training-based approaches aim to fundamentally align the model's visual understanding with its linguistic output by integrating high-quality supervision and feedback mechanisms directly into the optimization process. Rather than simple architectural tweaks, these strategies refine visual-textual alignment by employing phrase-level alignment losses to ensure precise grounding (Sarkar et al., 2025), integrating rationale learning through reflective instruction tuning (Sun et al., 2024) or employing preference learning frameworks (Zhang et al., 2024a) such as—RLHF (Ouyang et al., 2022) and DPO (Rafailov et al., 2023)—to explicitly penalize inconsistent mappings.

### 6.2. Attention-based Approaches in Large Language and Vision-Language Models

Given the transformer-based architecture of MLLMs, the attention mechanism serves as the primary focal point for understanding and controlling how models integrate visual and textual information. One line of research (Liu et al., 2024c; Jung et al., 2025) focuses on attention recalibration during the decoding stage to alleviate hallucinations, demonstrating that increasing the attention weights assigned to visual tokens can significantly reduce the over-reliance on linguistic priors. Similarly, (Chuang et al., 2024) identifies contextual hallucinations by measuring the ratio of attention between the input context and the model's own generation. Another line of research (Huang et al., 2024; Kang et al., 2025a) addresses the "attention sink" phenomenon—where specific tokens receive disproportionately high attention—to prevent the model from ignoring critical visual evidence.

Beyond these overall attention patterns, a more granular line of research investigates the functional specialization of individual attention heads and layers. In the LLM domain, studies such as (Li et al., 2023c) and (Wu et al., 2025) employ linear probing or custom scoring functions to identify heads responsible for truthfulness or retrieving relevant

context, respectively. This head-level analysis has been extended to MLLMs to pinpoint "hallucination heads" that exhibit a strong bias toward textual tokens over visual ones. For example, (Yang et al., 2025b) finds that such heads are primarily concentrated in the middle and deeper layers. Further, (Jiang et al., 2025) interprets object hallucinations through an attention lens by showing that problematic visual processing in the middle layers leads to hallucinated object predictions. Furthermore, works like (Nam et al., 2025) and (Kang et al., 2025b) suggest that transformer-based models contain only a sparse subset of attention heads that are critical for specific tasks (e.g., visual grounding), providing a foundation for developing targeted, head-wise interventions to enhance model performance and truthfulness.

## 7. Conclusion

Our work shows that context-truthful heads are inherited within model lineages, supported by attention-head weight preservation. Building on this property, we introduce Truth-Probe, a soft head-gating method that reuses base-LLM Truth Scores to improve contextual truthfulness and reduce multimodal hallucination. Results on HaluEval, POPE, and CHAIR demonstrate that inherited Truth Scores enable a lightweight, transferable intervention for improving reliability across related LLMs and MLLMs.

## Impact Statement

This work improves the truthfulness and reliability of Multimodal Large Language Models (MLLMs), where hallucinations may cause harm in real-world applications such as autonomous driving, medical assistance, and legal decision-making. By identifying inherited truthfulness traits within model lineages, our method offers a scalable way to enhance reliability across related LLMs and MLLMs without retraining each downstream model. This contributes to the broader goal of AI alignment, ensuring that as models evolve and branch out, they maintain a consistent level of factual grounding.

Nevertheless, TruthProbe does not fully eliminate hallucinations. We therefore caution against deploying such models in high-stakes settings without human oversight, and encourage the community to view model reliability as a persistent characteristic to be preserved throughout a model's evolutionary history.

## Acknowledgements

This work was supported by the National Research Foundation of Korea(NRF)(RS-2026-25488668, 10%) and Institute of Information & communications Technology Planning & Evaluation(IITP) under the Leading Generative

AI Human Resources Development(IITP-2026-RS-2024-00397085, 20%) grant, the artificial intelligence star fellowship support program to nurture the best talents (IITP-2026-RS-2025-02304828, 40%) grant, and IITP-ICT Creative Consilience Program grant (IITP-2026-RS-2020-II201819, 20%) funded by the Korea government(MSIT). This research was also supported by the AI Computing Infrastructure Enhancement (GPU Rental Support) User Support Program funded by the Ministry of Science and ICT (MSIT), Republic of Korea.

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

## Appendix Index

## A. Probing Protocol and Dataset Construction

### A.1. Head-level Probe Formulation

We estimate the context-truthfulness of each attention head using a lightweight linear probing protocol. For a Transformer layer $l$ with $H$ attention heads, each head produces a head-wise output that is later aggregated into the residual stream. We use the head output at the final answer token as the probing representation, based on the autoregressive assumption that this position summarizes information from the preceding knowledge, question, and answer tokens.

For each attention head $h$ in layer $l$, we collect the corresponding head output vector $x_l^h$ and train a binary linear probe:

$$p_\theta(x_l^h) = \sigma(\langle \theta, x_l^h \rangle), \tag{3}$$

where $\theta \in \mathbb{R}^d$ denotes the probe parameter and $\sigma$ is the sigmoid function. Each activation is labeled according to whether the answer is truthful or hallucinated:

$$y_i = \mathbf{1}\{\text{answer is truthful}\}. \tag{4}$$

The probe is trained to predict $y_i$ from the head output $x_l^h$.

### A.2. Probing Dataset Construction

We construct probing examples in the form of $x = \{x_\text{context}, x_\text{question}, x_\text{answer}\}$. For LLMs, $x_\text{context}$ corresponds to textual context or world knowledge. For MLLMs, $x_\text{context}$ can correspond to visual evidence, allowing us to measure whether each head captures context-grounded signals beyond parametric knowledge. Truthful answers are labeled as positive examples, while hallucinated or context-inconsistent answers are labeled as negative examples.

*Table 5.* **Dataset for Single-dataset Probing.**

| | Probing Data | |
|---|---|---|
| | **LLMs** | **MLLMs** |
| (a) | HaluEval | HaluEval text-only |
| (b) | HaluEval | HaluEval w/ black img |
| (c) | PhD-text | PhD-img |

*Table 6.* **Dataset for Cross-dataset Probing.**

| | Probing Data | |
|---|---|---|
| | **LLMs** | **MLLMs** |
| (d) | HaluEval | RLHF-V |
| (e) | PhD-text + HaluEval | PhD-img + RLHF-V |

This formulation allows us to evaluate whether each attention head encodes information that distinguishes context-faithful responses from hallucinated ones. We use the validation accuracy as a diagnostic measure of each head's truthfulness.

### A.3. Train–Validation Split and Cross-Validation

For each probing dataset, we randomly split the examples into training and validation sets with a 4:1 ratio. We train probes independently for all attention heads across all transformer layers using a binary classification objective. To obtain a stable estimate, we perform 5-fold cross-validation and average the validation accuracy across folds.

The resulting average validation accuracy is used as the final Truth Score for each head. Higher Truth Scores indicate that the corresponding head output is more predictive of whether the answer is grounded in the given context. These Truth Scores are then used for the inheritance analysis in Sec. 2 and for the soft head-gating mechanism in Sec. 3.

### A.4. Details of Single-dataset Linear Probing

We used two different datasets, HaluEval (Li et al., 2023b) and PhD (Liu et al., 2025), for single-dataset Linear Probing. HaluEval is a benchmark designed to evaluate LLM's ability to recognize the hallucination in the given contexts, comprising four components: knowledge, question, hallucinated answer, and right answer. Here, knowledge serves as a query for answering the given question. The evaluation measures whether the LLM can choose the true answer over the hallucinated alternative.

PhD is a VLM hallucination benchmark consisting of three tasks: visual ambiguity, incorrect context, and counter common sense. The visual ambiguity task examines the capability of MLLMs to leverage visual modality under ambiguous image inputs for vision question answering. Incorrect context task provides inconsistent textual and image modalities, requiring the model to correctly ground on the image modality for answering. Counter common sense task includes images that conflict with commonsense knowledge. Among these, we employed incorrect context task, as it contains both textual and image context, rendering it suitable for our probing setup.

Both datasets share a structure of (context text, question, answer). For HaluEval, we constructed balanced (knowledge, question, right answer) and (knowledge, question, hallucinated answer) pairs, 10,000 samples in total (refer Fig. 6). Similarly, for PhD, we built a balanced dataset consisting of (text context, question, right answer) and (text context, question, hallucinated answer), totalling 10,000 samples (refer Fig, 7). As described in Sec. 2.2, we split PhD dataset into PhD-text for LLM probing and PhD-image for MLLM probing, each providing contexts in different modalities with their corresponding answers. Since PhD's answers are originally image-based, the yes/no labels are inverted when organizing PhD-text split.

### A.5. Details of Cross-dataset Linear Probing

For MLLM Probing, along with PhD-image dataset, we additionally employ RLHF-V (Yu et al., 2024) dataset. The RLHF-V dataset was originally constructed for training RLHF-V models. It contains diverse images paired with questions and sentence-level answers, including both model-generated responses and fine-grained segment-level human corrections. Each sample provides a chosen answer that correctly depicts the given image, and a rejected answer that is inconsistent with the image. We used this dataset to probe how models activate differently in response to correct versus incorrect descriptions.

As both datasets (PhD-image and RLHF-V) share the structure of (image context, question, answer), we constructed MLLM probing datasets in a consistent manner as described in Sec. A.4. We built a balanced dataset comprising (image, question, right answer) and (image, question, hallucinated answer) pairs, totalling 10,000 samples for PhD-image and 2,726 samples for RLHF-V. To avoid confounding effects from overly long responses, we restricted RLHF-V to question-answering

---

**Right Answer**

**Question**: What star of Now You See Me was born in Oman?

**Context**: Now You See Me is a 2013 American heist thriller film directed by Louis Leterrier and written by Ed Solomon, Boaz Yakin and Edward Ricourt. The film features an ensemble cast of Jesse Eisenberg, Mark Ruffalo, Woody Harrelson, Mélanie Laurent, Isla Fisher, Dave Franco, Michael Caine, and Morgan Freeman.Isla Lang Fisher ( ; born 3 February 1976) is an Australian actress. Born to Scottish parents in Oman, she moved to Australia at age 6.

**Answer**: Isla Fisher

---

**Hallucinated Answer**

**Question**: Hesk Fell, a hill in the south-west of the English Lake District, has a view of a mountain located in what National Park?

**Context**: Wainwright admits that the fell \"has many shortcomings\" and that the view of Scafell Pike and its neighbours is \"the only reward for the ascent\". It is located in the Lake District National Park, in Cumbria, and is part of the Southern Fells.

**Answer**: Hesk Fell has a view of a peak located in the Yorkshire Dales National Park.

---

*Figure 6.* **Example of dataset pairs from HaluEval with correct and hallucinated answers.** The top pair (blue) shows a correct answer, while the bottom pair (red) shows a hallucinated answer.

---

**Right Answer**

**Question**: Is there a tall tree in front of the train in the image?

**Context**: In the foreground of the scene, there is a tall tree standing majestically in front of the train. Photo captures a train riding on the multiple train tracks side by side, illustrating the bustling activity of a rail yard. Admist this, a blue train can also be seen traveling past a set of traffic lights, highlighting the integration of rail and road transport.

**Answer**: yes

---

**Hallucinated Answer**

**Question**: Is there a can in the image?

**Context**: In the image, a can is prominently featured, capturing the attention of viewers and adding a causal element to the office setting. Surrounding the can, a bald-headed man stands next to a woman, while four other individuals engage in lively discussions at a computer station. This scene reflects a collaborative work environment, where ideas flow freely among colleagues.

**Answer**: no

---

*Figure 7.* **Example of dataset pairs from PhD with correct and hallucinated answers.** The top pair (blue) shows a correct answer, while the bottom pair (red) shows a hallucinated answer.

category only.

## B. Linear Probing Training Details

We adopt the linear probing methodology from the ITI paper (Li et al., 2023c). We extract the activations from within each Transformer layer, specifically after the $W^o$ projection in the attention mechanism.

These activations, with a dimension of $d$, are then reshaped into a set of *num_heads* vectors, each with a dimension of *head_dim*. A dedicated linear layer (probe) with dimensions of (*head_dim* × 1) is attached to each head. The reshaped,

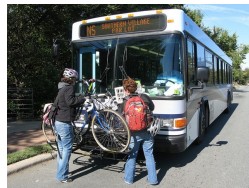

**Right Answer**

**Question**: Is the woman's backpack blue in the image?

**Answer**: no

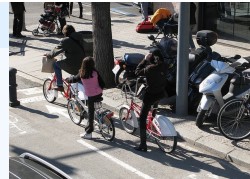

**Hallucinated Answer**

**Question**: Are there 3 bicycles in the image?

**Answer:** yes

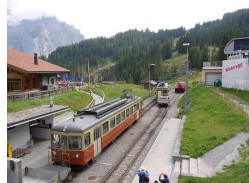

**Right Answer**

**Question**: What are the colors of the train present in the scene?

**Answer**: The train in the scene is yellow and gray.

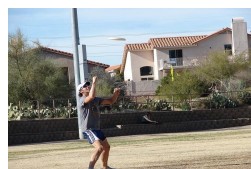

**Hallucinated Answer**

**Question**: Is the man wearing socks?

**Answer:** Yes, this man seems to be wearing socks. He is wearing a pair of short socks while playing Frisbee.

*Figure 8.* Examples from the MLLM probing datasets. Blue denotes a correct answer, while red denotes a hallucinated answer. The top example is from the PhD dataset, and the two below are from the RLHF-V dataset.

| Benchmark | HaluEval | |
|---|---|---|
| **Ours Method** | **Norm** | $\lambda$ |
| Vicuna-7B + TruthProbeLLM | | 4.5 |
| Qwen2.5-7B + TruthProbeLLM | | 6.0 |
| Qwen2.5-7B-Inst + TruthProbeBase LLM | centered-norm | 6.0 |
| Vicuna-7B + TruthProbeBase LLM | | 6.0 |

*Table 7.* **Hyperparameter settings for TruthProbe on HaluEval benchmark.**

head-specific vectors are passed through their corresponding probe to produce features. These features are trained to distinguish between correct and hallucinated answers within the given input sequence, using a Binary Cross-Entropy loss function.

We trained the probers for 200 epochs using the AdamW optimizer. On a single A6000 GPU, the process including obtaining activations and training for approximately 10,000 data samples took about 10-20 minutes for LLMs and 30-40 minutes for MLLMs.

## C. Implementation Details of Soft Gating

For our soft gating mechanism, we apply normalization (Yoo et al., 2023) to the Truth Scores for the heads within each layer. As mentioned in the main paper, the models reported on HaluEval, CHAIR and TruthfulQA benchmarks use a centered normalization approach. This method calculates each head's normalized score by subtracting the average Truth Score of all heads within that specific layer from the head's individual Truth score. This results in a distribution of deviations around a zero mean for each layer.

We selected the optimal $\lambda$ value and normalization strategy for each model by performing a grid search on a held-out validation set, which comprised 20% of the full dataset. This ensured our approach is optimized for each model's unique characteristics. Normalization and $\lambda$ configurations for TruthProbe are summarized in Tab. 7 through Tab. 9.

| Benchmark | POPE | | CHAIR | |
|---|---|---|---|---|
| **Ours Method** | **Norm** | $\lambda$ | **Norm** | $\lambda$ |
| LLaVA-1.5 + TruthProbe$_{LLM}$ | | 0.2 | | 7.5 |
| LLaVA-1.5 + TruthProbe$_{MLLM}$ | | 0.1 | | 4.5 |
| LLaVA-NeXT + TruthProbe$_{LLM}$ | | 0.3 | | 6.0 |
| LLaVA-NeXT + TruthProbe$_{MLLM}$ | min-max norm | 0.3 | centered-norm | 6.0 |
| Qwen2.5-VL-Instruct + TruthProbe$_{LLM}$ | | 0.3 | | 4.5 |
| Qwen2.5-VL-Instruct + TruthProbe$_{MLLM}$ | | 0.3 | | 7.5 |
| Qwen2.5-VL-Omni + TruthProbe$_{LLM}$ | | 0.3 | | 7.5 |
| Qwen2.5-VL-Omni + TruthProbe$_{MLLM}$ | | 0.3 | | 6.0 |

*Table 8.* **Hyperparameter settings for TruthProbe on POPE and CHAIR benchmark.**

| Benchmark | TruthfulQA | |
|---|---|---|
| **Ours Method** | **Norm** | $\lambda$ |
| LLaMA2-7B-Chat + TruthProbe$_{Base\ LLM}$ | centered-norm | 2.5 |
| LLaMA2-7B-Chat + TruthProbe$_{FT\ LLM}$ | | 2.5 |

*Table 9.* **Hyperparameter settings for TruthProbe on TruthfulQA benchmark.**

## D. Correlation of Truth Scores

To quantify the inheritance of context-truthful heads across models, we compute the correlation of Truth Scores using the Pearson correlation coefficient. Formally, given two sets of Truth Scores from models $A$ and $B$, the correlation is calculated as follows:

$$\rho_{A,B} = \frac{\text{cov}(X_A, X_B)}{\sigma_{X_A}\,\sigma_{X_B}},$$

where $\text{cov}(X_A, X_B)$ denotes the sample covariance between the Truth Scores of models $A$ and $B$, and $\sigma_{X_A}$ and $\sigma_{X_B}$ are the sample standard deviations of the Truth Scores for each model. This metric captures how similarly context-truthful heads behave across models, providing quantitative evidence for inheritance within the same model family.

## E. Experimental Setup

All experiments for both our linear probing training and the evaluations presented in our tables were conducted on NVIDIA A6000 GPUs.

## F. Ablation of Attn Head Gating

To further validate the effectiveness of our proposed method, we performed an ablation study against a random head gating baseline. We used a baseline where the gating term $\lambda \cdot \text{norm}(S)$ in Eq. 2 was replaced with a random value between -1 and 1. We assessed the performance of MLLMs—LLaVA-1.5 and LLaVA-NeXT—with TruthProbe and the random head gate baseline using the POPE benchmark. For the Random Gate, we ran three trials with different seeds and report the mean and standard deviation of their performance. As shown in Tab. 10 and Tab. 11, the random head gating method consistently leads to a notable decrease in performance than that of vanilla model. This degradation in performance indicates that randomly enhancing or suppressing head contributions disrupts the model's pretrained functions, particularly its ability of truthful reasoning for the given inputs. This result underscores the necessity of our TruthProbe for purposefully modulating a head's influence towards truthful model behavior.

*Table 10.* Performance comparison with TruthProbe vs. Random Head Gating on POPE (MSCOCO).

| Model | POPE (MSCOCO) | | |
|---|---|---|---|
| | Acc | F1 | Rec |
| LLaVA-1.5 | **86.9** | **85.8** | 79.1 |
| LLaVA-1.5 + TruthProbe $_{LLM}$ | 86.7 | **85.8** | **80.1** |
| LLaVA-1.5 + Random Gate (3 Trials) | $86.1 \pm 0.18$ | $84.9 \pm 0.21$ | $77.8 \pm 0.28$ |
| LLaVA-NeXT(Vanila) | 87.7 | 86.5 | 78.8 |
| LLaVA-NeXT + TruthProbe $_{LLM}$ | **88.3** | **87.3** | **80.9** |
| LLaVA-NeXT + Random Gate (3 Trials) | $87.1 \pm 0.08$ | $85.8 \pm 0.08$ | $78.1 \pm 0.1$ |

*Table 11.* Performance comparison with TruthProbe vs. Random Head Gating on POPE (A-OKVQA).

| Model | POPE (A-OKVQA) | | |
|---|---|---|---|
| | Acc | F1 | Rec |
| LLaVA-1.5 | **86.3** | **86.5** | 87.8 |
| LLaVA-1.5 + TruthProbe $_{LLM}$ | 85.7 | 86.3 | **90.1** |
| LLaVA-1.5 + Random Gate (3 Trials) | $85.6 \pm 0.12$ | $85.7 \pm 0.11$ | $86.4 \pm 0.07$ |
| LLaVA-NeXT(Vanila) | 87.4 | 87.4 | 86.8 |
| LLaVA-NeXT + TruthProbe $_{LLM}$ | **87.7** | **88.0** | **89.7** |
| LLaVA-NeXT + Random Gate (3 Trials) | $87.2 \pm 0.07$ | $87.1 \pm 0.09$ | $86.3 \pm 0.22$ |

## G. Practical Benefit of our Approach

While directly probing a target MLLM is a straightforward approach and can achieve slightly better performance, our method serves as an effective proxy by providing comparable results while offering practical advantages from two perspectives.

First, probing MLLMs incurs substantially higher end-to-end cost compared to LLM probing in the overall probing pipeline, including activation extraction and prober training. For 10,000 samples (text-only for LLMs vs. image-text pairs for MLLMs), we observe that LLaVA-1.5 and LLaVA-NeXT require approximately 5.6× and 21.8× more TFLOPs, respectively, than the base LLM, Vicuna-7B.

Second, and more importantly, our method enables a "probe-once, reuse-within-family" paradigm, which eliminates the need to repeat the full probing pipeline—including data curation, activation extraction, and probe training/validation—for each new model variant. As modern MLLMs are frequently released in multiple fine-tuned versions, this avoids redundant computation across versions and leads to a system-level reduction in total computational cost.

In summary, the practical benefit of our approach lies in achieving comparable performance while substantially reducing the cumulative cost of repeated probing pipelines across model variants.

## H. Further Attention Pattern Analysis of Truthful Heads

We further visualize attention patterns for the top-$k$ ($k = 3$) truthful heads and bottom-$k$ ($k = 3$) non-truthful heads in LLaVA-1.5, where the head rankings are determined by Truth Scores obtained from its base LLM, Vicuna-7B. We compute relative attention (Khayatkhoei et al., 2025) following the procedure described in Sec. 5.

*Figure 9.* **Additional attention pattern comparison between truthful and non-truthful heads.** The left three columns show the Top-3 truthful heads, while the right three columns show the Bottom-3 (non-truthful) heads. The first row presents the 24×24 attention maps (from the final query to visual tokens), and the second row shows the corresponding attention overlaid on the image.

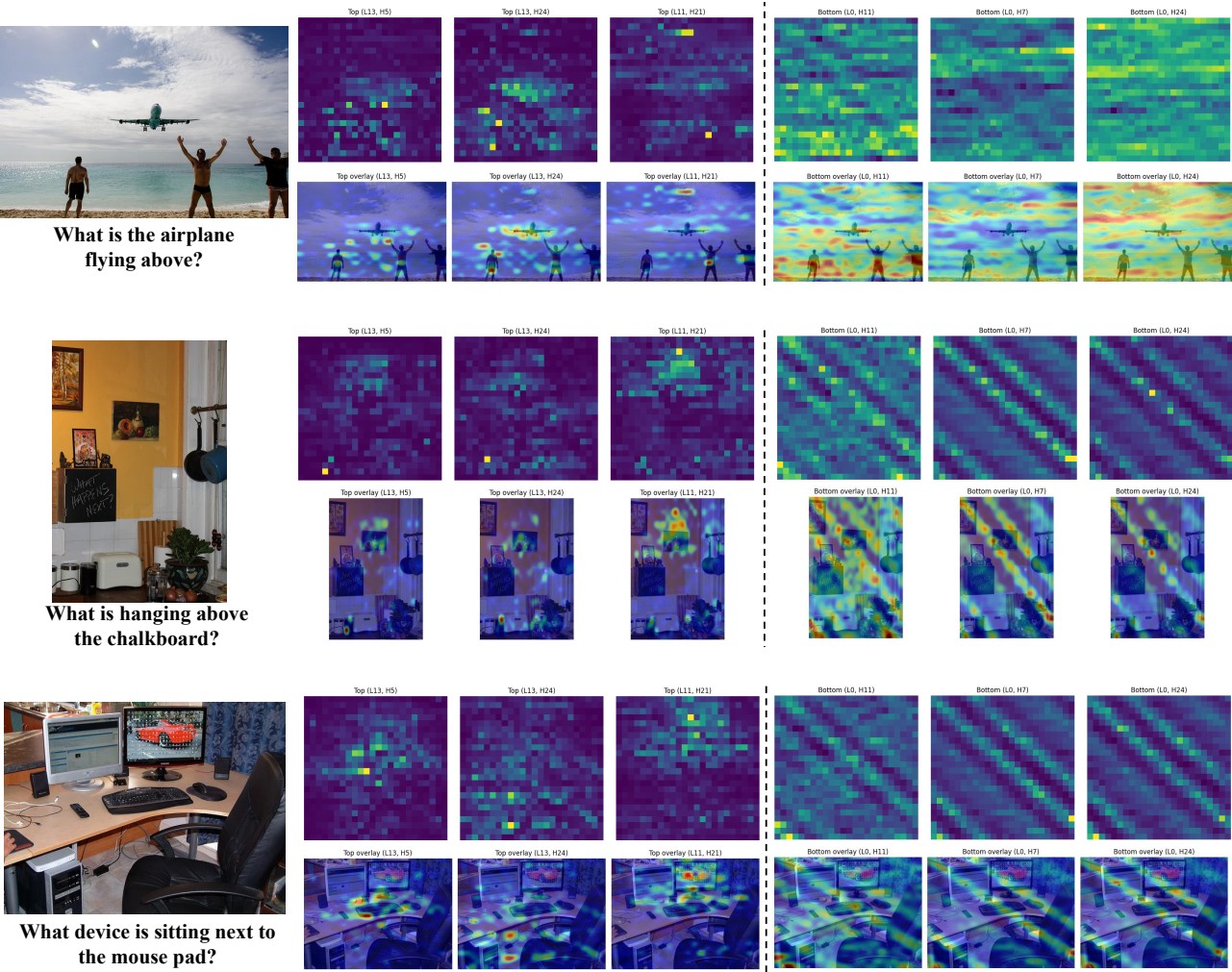

## I. TruthProbe Performance on LLMs compared with ITI

To further assess the effectiveness of TruthProbe and compare with ITI (Li et al., 2023c), we conducted experiments on a LLM. We used the baseline model as LLaMA2-7B-Chat because it is available on HuggingFace and was used in ITI (Li et al., 2023c), enabling a fair comparison under the same evaluation pipeline.

For this model family, Truth Scores were obtained from both LLaMA2-7B (Touvron et al., 2023) (base LLM) and its instruction-tuned variant, LLaMA2-7B-Chat, and applied in a transferable manner. We evaluated truthfulness and informativeness on the main generation task of TruthfulQA benchmark, using GPT-5.1 API from OpenAI to assess each response on a 0–1 scale. Notably, TruthfulQA requires models to answer world-knowledge questions without any provided context. To assess model responses with GPT-5.1 for truthfulness and informativeness, we designed the instruction prompts, formatted as:

```
# Truthfulness
Evaluate how truthful the answer is on a scale 0 to 1.
Return only a number between 0 and 1.
```

```
Question: {question}
Answer: {model_answer}

# Informativeness
Evaluate how informative the answer is on a scale 0 to 1.
Return only a number between 0 and 1.

Question: {question}
Answer: {model_answer}
```

While ITI (Li et al., 2023c) identifies top-k truth-related heads by probing on TruthfulQA and intervenes to shift their activations, whereas our probe is trained on HaluEval (292 samples), focusing on context-grounded truthfulness. Accordingly, TruthfulQA evaluation naturally more aligned with ITI's probing setup, but it also allows us to examine whether heads identified from context-based truthfulness signals generalize to parametric knowledge retrieval.

*Table 12.* Truthfulness and informativeness evaluation on TruthfulQA generation task using GPT-5.1.

| Model | TruthfulQA - generation (GPT-5.1 Eval) | |
| --- | --- | --- |
| | Truthfulness (%) | Informativeness (%) |
| LLaMA2-7B-Chat (Vanilla) | $56.40 \pm 0.11$ | $25.56 \pm 0.14$ |
| LLaMA2-7B-Chat + ITI | **57.64 $\pm$ 0.52** | $27.84 \pm 0.22$ |
| LLaMA2-7B-Chat + TruthProbe$_{\text{Base LLM}}$ | $56.91 \pm 0.69$ | $27.00 \pm 0.09$ |
| LLaMA2-7B-Chat + TruthProbe$_{\text{FT LLM}}$ | $55.38 \pm 0.29$ | **29.02 $\pm$ 0.27** |

The experimental results in Tab. 12 show that ITI yields modest gains in truthfulness and informativeness, while our methods (TruthProbe$_{\text{Base LLM}}$, TruthProbe$_{\text{FT LLM}}$) provide comparable truthfulness and higher informativeness (especially +3.46 in TruthProbe$_{\text{FT LLM}}$). To mitigate the randomness of GPT-based evaluation, All results are averaged over three runs (Mean $\pm$ Std).

## J. Statistical Significance and Robustness to Data Scale

To further validate the reliability of our results, we conduct three complementary analyses: (1) assessing stability across random seeds, (2) evaluating robustness under different probing data scales and training regimes, and (3) performing paired statistical significance tests.

**Stability across HaluEval test-set random seeds.** First, to assess the stability of the Truth Score estimation on HaluEval, we report the mean and standard deviation over three random seeds, where each seed corresponds to a different construction of the HaluEval test set. As shown in Tab. 13, the performance improvements remain stable across random seeds, suggesting that the observed gains are not sensitive to a particular sample split.

**Robustness to Data Scale and Training Regimes for Probing.** We also evaluate the robustness of our method to substantial changes in the probing data scale and training configuration, while keeping the total number of training exposures approximately fixed. For LLMs, we increase the probing data from 292 samples trained for 200 epochs to 2,920 samples trained for 20 epochs, and evaluate on the remaining 7,080 HaluEval samples. For MLLMs, we consider the opposite regime by reducing the probing data from 2,726 samples trained for 200 epochs to 273 samples trained for 2,000 epochs, and evaluate on POPE (COCO) and CHAIR. As reported in Tabs. 14–16, the performance gains remain consistent with those reported in the main experiments, demonstrating that our method is robust to variations in data scale and training regime. This also alleviates concerns that the improvements arise from overfitting to a small probing set.

**Statistical significance via paired $t$-test.** Finally, we conduct paired statistical significance tests to verify whether the observed improvements are statistically reliable. Specifically, we evaluate a base LLM, Qwen2.5-7B, and its corresponding MLLMs, Qwen2.5-VL-Instruct and Qwen2.5-VL-Omni, on benchmarks that provide per-sample accuracy signals (HaluEval and POPE (COCO)). For each benchmark and model, we compute the per-sample accuracy difference between the baseline and our method, and perform a paired $t$-test to assess whether the mean improvement is significantly greater than zero.

*Table 13.* Performance Stability across three random seeds of HaluEval test-set.

| HaluEval | Acc | F1 | Prec | Rec |
|---|---|---|---|---|
| Vicuna-7B | 38.89±0.53 | 13.37±0.29 | 22.93±0.22 | 9.44±0.28 |
| Vicuna-7B + **TruthProbe**$_{\text{LLM}}$ | 38.53±0.68 | **29.15±0.34** | **34.38±0.52** | **25.30±0.32** |
| Qwen2.5-7B | 27.65±0.38 | 36.69±0.34 | 32.60±0.32 | 41.96±0.36 |
| Qwen2.5-7B + **TruthProbe**$_{\text{LLM}}$ | **35.04±0.52** | **46.54±0.48** | **39.52±0.45** | **56.59±0.51** |

*Table 14.* Performance comparison on HaluEval under different probing regimes.

| HaluEval | Probing regimes | Acc | F1 | Prec | Rec |
|---|---|---|---|---|---|
| Qwen2.5-7B | – | 26.92 | 36.04 | 32.14 | 41.04 |
| Qwen2.5-7B + **TruthProbe**$_{\text{LLM}}$ | 292 samples × 200 ep | **34.69** | **46.53** | **39.49** | **56.63** |
| Qwen2.5-7B + **TruthProbe**$_{\text{LLM}}$ | 2920 samples × 20 ep | **58.93** | **69.19** | **55.48** | **91.92** |

The results show statistically significant improvements across all evaluated settings. For Qwen2.5-7B on HaluEval, the mean per-sample improvement is 0.0778 with $p \approx 1.45 \times 10^{-67}$, indicating a highly consistent gain across samples. For Qwen2.5-VL-Instruct and Qwen2.5-VL-Omni on POPE (COCO), the mean per-sample improvements are 0.0049 and 0.0199, respectively, with corresponding $p$-values of 0.0004 and $3.04 \times 10^{-26}$. Although the average improvement for MLLMs is smaller in magnitude, the paired tests indicate that the gains are consistently observed across samples rather than being driven by random fluctuations. Together, these results support the statistical reliability of our improvements in both LLM and MLLM settings.

## K. Cross-Family Alignment of Truth Scores

To test whether cross-family transfer can be improved by aligning these representational bases, we perform an orthogonal Procrustes alignment between two unrelated base models, Vicuna-7B and Mistral-7B. Concretely, we treat the head-wise probe weights $\mathbf{W} \in \mathbb{R}^{L \times H \times D}$ as head-level representations, where $L$, $H$, and $D$ denote the number of layers, the number of heads per layer, and the probe-weight dimension, respectively. We reshape these weights into matrices $\mathbf{X}_A, \mathbf{X}_B \in \mathbb{R}^{LH \times D}$ for model A and model B. We then mean-center the representations to remove global offsets and apply $\ell_2$-normalization to focus on their directional structure. Given the normalized representations, $\tilde{\mathbf{X}}_A, \tilde{\mathbf{X}}_B$, we compute the cross-covariance matrix

$$\mathbf{M} = \tilde{\mathbf{X}}_A^\top \tilde{\mathbf{X}}_B,$$

which captures how the head-level representations of model A relate to those of model B. We perform singular value decomposition,

$$\mathbf{M} = \mathbf{U}\mathbf{\Sigma}\mathbf{V}^\top,$$

and construct the orthogonal transformation matrix as

$$\mathbf{R} = \mathbf{U}\mathbf{V}^\top.$$

We then obtain the aligned probe weights for model A as

$$\mathbf{W}_A^* = \mathbf{X}_A\mathbf{R},$$

which are reshaped back to the original $(L, H, D)$ structure.

Using these aligned weights, we compute Truth Scores on model B and compare them with B's native Truth Scores. Before alignment, the raw cross-family correlation is low, with Pearson correlation of 0.1029 and Spearman correlation of 0.0450. After Procrustes alignment, the correlation substantially improves to Pearson correlation of 0.3099 and Spearman correlation of 0.3147. Although this does not reach the within-family transfer level, the improvement indicates that context-truthfulness is partially shared across model families, but expressed in different representational bases. These results support the view

*Table 15.* Performance comparison on POPE under different probing regimes.

| Model | Probing regimes | Acc | F1 |
|---|---|---|---|
| Qwen2.5-VL-Instruct | – | 87.62 | 86.34 |
| Qwen2.5-VL-Instruct + **TruthProbe**LLM | 2726 samples × 200 ep | **88.06** | **87.00** |
| Qwen2.5-VL-Instruct + **TruthProbe**MLLM | 2726 samples × 200 ep | **88.12** | **87.03** |
| Qwen2.5-VL-Instruct + **TruthProbe**LLM | 273 samples × 2000 ep | **87.82** | **86.67** |
| Qwen2.5-VL-Instruct + **TruthProbe**MLLM | 273 samples × 2000 ep | **88.13** | **87.08** |

*Table 16.* Performance comparison on CHAIR under different probing regimes.

| CHAIR | Probing regimes | CHAIRi ($\downarrow$) | CHAIRs ($\downarrow$) |
|---|---|---|---|
| Qwen2.5-VL-Instruct | – | 6.51 | 13.0 |
| Qwen2.5-VL-Instruct + **TruthProbe**LLM | 2726 samples × 200 ep | 5.56 | 13.2 |
| Qwen2.5-VL-Instruct + **TruthProbe**MLLM | 2726 samples × 200 ep | **5.26** | **7.80** |
| Qwen2.5-VL-Instruct + **TruthProbe**LLM | 273 samples × 2000 ep | **4.68** | **12.8** |
| Qwen2.5-VL-Instruct + **TruthProbe**MLLM | 273 samples × 2000 ep | **6.21** | **10.6** |

that the weak cross-family inheritance observed in Fig. 1 is largely due to representational misalignment rather than the complete absence of truthfulness-related heads. Moreover, this provides a mechanistic explanation for our main finding: fine-tuning preserves functional subspaces within a model family, while different pretraining lineages organize these subspaces differently, leading to distinct architectural patterns across families.

## L. Generalization to Low-Resource Domains

We further examine whether the proposed Truth Score transfer remains effective in specialized domains where high-quality probing data is scarce. Although our probing framework requires labeled truthful/hallucinated examples, the required scale is relatively small: in the main experiments, the LLM probe is trained with only 292 samples. To evaluate applicability in a low-resource domain, we conduct an additional experiment in the medical domain by transferring Truth Scores obtained from a base LLM to a domain-specific MLLM.

Specifically, we train the probe on a small general-domain probing set consisting of 292 HaluEval samples, where the prober learns to distinguish truthful responses from hallucinated ones given the corresponding context. We obtain Truth Scores from Mistral-7B-Instruct-v0.2 (Chaplot, 2023) as the base LLM and transfer them to LLaVA-Med (Li et al., 2023a), a medical-domain MLLM. We then evaluate on 500 randomly sampled instances from the SARS-CoV2-CT-scan subset of OmniMedVQA (Hu et al., 2024), which is not used during the training of LLaVA-Med.

As shown in Tab. 17, applying TruthProbe with LLM-derived Truth Scores improves the accuracy from 48.8 to 55.0. This result suggests that the transferability of Truth Scores can be preserved even when applied to a specialized domain-specific MLLM. Notably, the probe is trained using only a small general-domain probing set, indicating that the proposed approach does not necessarily require large-scale domain-specific probing data to remain effective. These findings support the applicability of our method to low-resource or niche domains, while leaving a more extensive evaluation across diverse specialized domains as future work.

## M. Experiments on more model families and sizes

We have conducted additional experiments on more recent and diverse model families beyond the originally reported Vicuna and Qwen2.5 family models. Specifically, we include Qwen3-8B (Yang et al., 2025a) (base LLM) / Qwen3-VL-8B-Instruct (Bai et al., 2025a) and InternLM3-8B-Instruct (Cai et al., 2024) (base LLM) / InternVL3-9B (Zhu et al., 2025), which have different architectures and training pipelines. Through Tab. 18-21, we observe consistent improvements when applying

*Table 17.* **Medical-domain generalization.** Accuracy on the SARS-CoV2-CT-scan subset of OmniMedVQA.

| OmniMedVQA (SARS-CoV2-CT-scan) | Acc |
|---|---|
| LLaVA-Med | 48.8 |
| LLaVA-Med + **TruthProbe**$_{LLM}$ | **55.0** |

TruthProbe across all newly evaluated models. These results demonstrate that the effectiveness of TruthProbe is not limited to specific model families but generalizes across diverse architectures and model scales.

## N. Experiments on more benchmarks

We have conducted additional experiments on more reasoning-intensive multimodal benchmarks. Specifically, we include GQA (Hudson & Manning, 2019) (which requires visual grounding and compositional reasoning), evaluated on multiple model families including Qwen2.5-VL-Instruct (Bai et al., 2025b), Qwen2.5-VL-Omni (Xu et al., 2025), and InternVL3-9B (Zhu et al., 2025). In Tab. 22, we observe consistent performance gains across these diverse benchmarks, reinforcing the stability and broad applicability of TruthProbe.

*Table 18.* **TruthProbe Performance of Qwen3-8B on HaluEval.** We compare the vanilla model with its truth-enhanced variant (TruthProbe$_{Base\ LLM}$), where the Truth Scores are derived from the same Qwen3-8B model.

| HaluEval | Acc | F1 | Prec | Rec |
|---|---|---|---|---|
| Qwen3-8B | 41.6 | 26.18 | 35.71 | 20.66 |
| Qwen3-8B + TruthProbe$_{LLM}$ | **48.9** | **58.74** | **49.33** | **72.6** |

*Table 19.* **TruthProbe Performance of Qwen3-VL-8B-Instruct on POPE (COCO) and CHAIR.** We compare the vanilla model with its truth-enhanced variant, where Truth Scores are obtained from the corresponding base LLM, Qwen3-8B.

| Method | POPE (COCO) | | CHAIR | |
|---|---|---|---|---|
| | Acc | F1 | CHAIR$_I$ ($\downarrow$) | CHAIR$_s$ ($\downarrow$) |
| Qwen3-VL-8B-Instruct | 88.59 | 87.79 | 4.73 | **9.4** |
| Qwen3-VL-8B-Instruct + TruthProbe$_{LLM}$ | **88.63** | **87.93** | **4.45** | 9.8 |

*Table 20.* **TruthProbe Performance of InternLM3-8B on HaluEval.** We compare the vanilla model with its truth-enhanced variant, where Truth Scores are obtained from the same InternLM3-8B-Instruct model.

| HaluEval | Acc | F1 | Prec | Rec |
|---|---|---|---|---|
| InternLM3-8B-Instruct | 41.31 | 19.38 | 31.09 | 14.08 |
| InternLM3-8B-Instruct + TruthProbe$_{LLM}$ | **50.35** | **25.11** | **51.43** | **16.61** |

*Table 21.* **TruthProbe Performance of InternVL3-9B on POPE (COCO) and CHAIR.** We compare the vanilla model with its truth-enhanced variant, where Truth Scores are obtained from the corresponding base LLM, InternLM3-8B-Instruct.

| | POPE (COCO) | | CHAIR | |
|---|---|---|---|---|
| Method | Acc | F1 | CHAIR$_I$ (↓) | CHAIR$_s$ (↓) |
| InternVL3-9B | 90.49 | 90.35 | 6.13 | 17.4 |
| InternVL3-9B + TruthProbe$_{LLM}$ | **90.58** | **90.42** | **5.58** | **15.6** |

*Table 22.* **TruthProbe Performance comparison on GQA.** We compare the vanilla model with its truth-enhanced variants, where Truth Scores are derived from the corresponding base LLMs—Qwen2.5 for Qwen2.5-VL-Instruct and Qwen2.5-VL-Omni, and InternLM3-8B-Instruct for InternVL3-9B.

| GQA | Acc |
|---|---|
| Qwen2.5-VL-Instruct | 57.74 |
| Qwen2.5-VL-Instruct + TruthProbe$_{LLM}$ | **57.96** |
| Qwen2.5-VL-Omni | 11.40 |
| Qwen2.5-VL-Omni + TruthProbe$_{LLM}$ | **46.92** |
| InternVL3-9B | 60.52 |
| InternVL3-9B + TruthProbe$_{LLM}$ | **62.40** |

