# OpenReview forum: "The Truth Stays in the Family: Enhancing Contextual Truthfulness via Inherited Heads in Model Lineages"
_ICML.cc/2026/Conference — ICML 2026 regular_

### Official Review · Reviewer_4Zjf · 2026-02-26

**Soundness:** 3
**Presentation:** 3
**Significance:** 3
**Originality:** 3
**Overall Recommendation:** 4
**Confidence:** 3

**Summary:**

This paper studies whether context-truthful attention heads are preserved across model lineages from foundational LLMs to their fine-tuned LLM/MLLM descendants. The authors introduce a head-level linear probing approach to quantify “Context-Truthfulness Scores” and report strong correlations of these scores within model families (e.g., Vicuna → LLaVA, Qwen → Qwen-VL variants). Based on this inheritance phenomenon, they propose a lightweight soft head-gating mechanism (TruthProbe) that amplifies high-truthfulness heads during inference, improving performance on HaluEval, POPE, and CHAIR benchmarks.

The central claim is that truthfulness-related components are structurally inherited along model lineages and can be exploited in a plug-and-play manner to improve reliability across a family of models.

**Compliance With Llm Reviewing Policy:**

Affirmed.

**Final Justification:**

The author has addressed my concern

**Key Questions For Authors:**

1. The models used in the experiments are relatively small in scale, and the paper does not demonstrate whether the proposed idea holds for models larger than 7B parameters. Moreover, the authors do not evaluate more recent models such as Qwen3-VL. The selected multimodal models (e.g., LLaVA-1.5 and LLaVA-NeXT) are somewhat outdated.

2. The experimental scope is limited, and the chosen benchmarks are relatively old. In particular, CHAIR dates back to 2018 and was originally designed for evaluating object hallucination in image captioning. It is somewhat inconsistent that the paper claims to investigate visual grounding in multimodal models while also relying on text-based hallucination benchmarks to validate the method, which feels conceptually misaligned.

**Strengths And Weaknesses:**

#Strengths

1. Interesting high-level hypothesis.
The idea of examining model reliability from a lineage-level perspective (rather than per-model fixes) is conceptually appealing and relatively underexplored.

2. Head-level interpretability analysis.
The paper leverages linear probing to identify context-truthful heads and performs both single-dataset and cross-dataset correlation analyses.

3. Transferability insight.
The empirical observation that Truth Scores from base LLMs can improve fine-tuned LLMs and MLLMs is intriguing and practically appealing.

4. Training-free intervention.
The proposed soft gating mechanism is lightweight and does not require retraining, which is practically useful.

# Weaknesses

1. The models used in the experiments are relatively small in scale, and the paper does not demonstrate whether the proposed idea holds for models larger than 7B parameters. Moreover, the authors do not evaluate more recent models such as Qwen3-VL. The selected multimodal models (e.g., LLaVA-1.5 and LLaVA-NeXT) are somewhat outdated.

2. The experimental scope is limited, and the chosen benchmarks are relatively old. In particular, CHAIR dates back to 2018 and was originally designed for evaluating object hallucination in image captioning. It is somewhat inconsistent that the paper claims to investigate visual grounding in multimodal models while also relying on text-based hallucination benchmarks to validate the method, which feels conceptually misaligned.

---

> ### Author Rebuttal · Authors · 2026-03-31
>
> We thank the reviewer for the valuable feedback regarding the experimental scope.
>
> **More model families and sizes**
>
> To address this, we have conducted additional experiments on more recent and diverse model families beyond the originally reported Vicuna and Qwen2.5 family models. Specifically, we include **Qwen3-8B** (base LLM) / **Qwen3-VL-8B-Instruct** and **InternLM3-8B-Instruct** (base LLM) / **InternVL3-9B**, which have different architectures and training pipelines.
> We observe consistent improvements when applying TruthProbe across all newly evaluated models. These results demonstrate that the effectiveness of TruthProbe is not limited to specific model families but generalizes across diverse architectures and model scales.
>
> **1) Qwen3-8B / Qwen3-VL-8B-Instruct**
>
> | HaluEval | Acc | F1 | Prec | Rec |
> |---|---|---|---|---|
> | Qwen3-8B | 41.60 | 26.18 | 35.71 | 20.66 |
> | Qwen3-8B + TruthProbe_LLM | **48.90** | **58.74** | **49.33** | **72.60** |
>
> **2) InternLM3-8B / InternVL3-9B**
>
> | HaluEval | Acc | F1 | Prec | Rec |
> |---|---|---|---|---|
> | InternLM3-8B | 41.31 | 19.38 | 31.09 | 14.08 |
> | InternLM3-8B + TruthProbe_LLM | **50.35** | **25.11** | **51.43** | **16.61** |
>
> | CHAIR | CHAIRi (↓) | CHAIRs (↓) |
> |---|---|---|
> | InternVL3-9B | 6.13 | 17.4 |
> | InternVL3-9B + TruthProbe_LLM | **5.58** | **15.6** |
>
> **More benchmarks**
> To address this concern, we have conducted additional experiments on more reasoning-intensive multimodal benchmarks. Specifically, we include **GQA** (which requires visual grounding and compositional reasoning) and **MMMU** (which evaluates complex multi-modal reasoning), both of which demand strong multimodal understanding, evaluated on multiple model families beyond those in the main paper.
> We observe consistent performance gains across these diverse benchmarks, reinforcing the stability and broad applicability of TruthProbe.
>
> **1) GQA**
> | GQA | Acc |
> |---|---|
> | Qwen2.5-VL-Instruct | 57.74 |
> | Qwen2.5-VL-Instruct + TruthProbe_LLM | **57.96** |
> | Qwen2.5-VL-Omni | 11.40 |
> | Qwen2.5-VL-Omni + TruthProbe_LLM | **46.92** |
> | InternVL3-9B | 60.52 |
> | InternVL3-9B + TruthProbe_LLM | **62.40** |
>
> **2) MMMU (Val) Math**
> | MMMU (Val) Math | Acc |
> |---|---|
> | Qwen2.5-VL-Omni | 36.67 |
> | Qwen2.5-VL-Omni + TruthProbe_LLM | **40.00** |

---

> > ### Author Rebuttal · Reviewer_4Zjf · 2026-04-01
> >
> > The author has addressed my concerns

---

> > > ### Author Response · Authors · 2026-04-02
> > >
> > > We sincerely thank the reviewer for the valuable feedback regarding the experimental scope. Addressing these points has significantly strengthened the empirical validation of our work.
> > >
> > > We appreciate for raising the score and are glad that our rebuttal has addressed your concerns. All newly added results will be fully incorporated into the final version.

---

### Official Review · Reviewer_ym5Q · 2026-03-08

**Soundness:** 2
**Presentation:** 2
**Significance:** 1
**Originality:** 2
**Overall Recommendation:** 3
**Confidence:** 3

**Summary:**

Overall, the authors outline a central concept: truthfulness-related traits of attention heads in base large language models (LLMs) are inherited by their fine-tuned multimodal descendants (MLLMs). The authors explore an important concept of leveraging this inheritance to enhance contextual truthfulness across model lineages. The work quantifies head-level "Truth Scores" via linear probing, reveals strong correlations between base LLMs and MLLMs (even across modalities/datasets), and proposes a Soft Gating strategy to amplify truthful heads. Experiments on HaluEval, POPE, and CHAIR show base LLM-derived Truth Scores achieve comparable gains to probing MLLMs directly, offering a plug-and-play solution for reducing hallucinations.

**Compliance With Llm Reviewing Policy:**

Affirmed.

**Final Justification:**

I appreciate your response and supplementary experiments. However, my key concern about insufficient empirical validation persists, so I raise the score a little bit.

**Key Questions For Authors:**

1. The inheritance of Truth Scores is strong within model families but negligible across unrelated models (Fig. 1). Do truthful heads in different lineages encode context-truthfulness through distinct architectural patterns (e.g., layer distribution, attention weights), and could cross-family transfer be enabled by aligning these patterns?
2. For niche domains (e.g., medical imaging, legal text) where high-quality probing datasets are scarce, how well would the method generalize? Would a small-domain adaptation set for probing be sufficient to maintain the transferability of Truth Scores from base LLMs to domain-specific MLLMs?
3. The Soft Gating uses a fixed λ for all layers (Tab. 7-8). Given that attention heads in different layers often have specialized functions (e.g., lower layers for syntax, deeper layers for semantics), would adaptive λ values per layer further improve truthfulness by better balancing the contributions of layer-specific truthful heads?

**Limitations:**

N.A

**Strengths And Weaknesses:**

Strengths
1. This work identifies that context-truthful attention heads are preserved across model lineages, even after multimodal fine-tuning or instruction tuning. This moves beyond isolated model fixes to a systemic approach for enhancing reliability across entire families.
2. The Soft Gating mechanism requires no model retraining, only modulation of attention head contributions based on precomputed Truth Scores. This plug-and-play design works across LLMs and MLLMs, eliminating the need for modality-specific tuning.

Weaknesses
1. The Truth Scores rely on high-quality labeled data (truthful vs. hallucinated pairs) for linear probing. For niche domains or low-resource languages, constructing such datasets may be challenging, limiting applicability.
2. The Soft Gating’s effectiveness depends on normalization strategy and λ scaling parameter, which require grid search optimization for each model/benchmark. This adds complexity to deployment across diverse model variants.
3. The work confirms inheritance but does not explore how fine-tuning modifies the specific functions of truthful heads (e.g., whether they adapt to visual context in MLLMs), leaving the mechanism of inheritance poorly understood.

---

> ### Author Rebuttal · Authors · 2026-03-31
>
> We thank the reviewer for the insightful suggestion on generalization to low-resource domains, mechanism of inheritance, cross-family alignment, and the design of the soft gating mechanism.
>
> > W1 & Q2 (generalization to low-resource domains)
>
> While our probing framework does rely on labeled data (truthful vs. hallucinated), the required scale is relatively small—in our main experiments, the LLM probe is trained with only 292 samples.
> To further address the concern about applicability in low-resource or niche domains, we conduct an additional experiment in the medical domain, where high-quality labeled data is typically scarce. Specifically, we obtain Truth Scores from a small probing set (292 HaluEval samples, from which prober learns to discern truthfulness of given general context) using Mistral-7B-Instruct-v0.2 (Base LLM) and transfer them to a domain-specific MLLM, LLaVA-Med. We evaluate on 500 randomly sampled instances from the SARS-CoV2-CT-scan subset of the OmniMedVQA (not used during training of LLaVA-Med).
> The baseline accuracy is 48.8, while applying TruthProbe (LLM-based) improves performance to 55.0, demonstrating that transferability is preserved even in a specialized domain with limited probing data. These results suggest that a small domain-specific probing set is sufficient to maintain effectiveness, alleviating concerns about scalability to low-resource settings.
>
> | OmniMedVQA (SARS-CoV2-CT-scan) | Acc |
> |---|---|
> | LLaVA-Med | 48.8 |
> | LLaVA-Med + TruthProbe_LLM | **55.0** |
>
> > W3 (mechanism of inheritance)
>
> Our analysis shows that truthful heads are primarily located in middle-to-deep layers that undergo minimal modification during fine-tuning. Please refer to our response to Reviewer WFS6 for the detailed results.
>
> > Q1 (cross-family alignment)
>
> We thank the reviewer for this insightful question.
>
> To address whether truthful heads across different lineages encode context-truthfulness through distinct architectural patterns, we analyze the degree of weight drift within-family and cross-family (Fig.2 in [Link](https://ibb.co/s9NRPqJS)). We observe a clear contrast: weight drift is minimal within the same family but larger across different families, indicating that the underlying representations are organized differently. These results suggest that truthful heads are indeed realized through distinct architectural patterns across model lineages.
>
> To investigate whether cross-family transfer can be enabled by aligning architectural patterns, we perform a Procrustes alignment between two unrelated base models, Vicuna-7B (model A) and Mistral-7B (model B).
>
> Concretely, we treat head-wise probe weights $W\in R^{L\times H\times D}$ as representations, reshape them into a matrix of size (LH,D).
> We then apply (1) mean-centering to remove global offsets, (2) L2-normalization to focus on directional structure.
> Next, we apply (3) orthogonal Procrustes alignment.
> We compute a cross-covariance matrix, $M = X_A^⊤ X_B$, which captures how representations from model A relate to those from model B.
> We perform SVD on this matrix, i.e. $M=UΣV^⊤$, and construct an orthogonal transformation matrix as $R=UV^⊤$, from which we obtain aligned prober weights $W_A^∗​=X_A ​R$ reshaped back to (L, H, D).
>
> Using these aligned weights, we compute Truth Scores on model B and compare them with B’s native scores. While the raw cross-family correlation is low (Pearson: 0.1029, Spearman: 0.0450), alignment significantly improves it (Pearson: 0.3099, Spearman: 0.3147).
> Although this does not reach within-family levels, the substantial gain indicates that truthfulness is partially shared across model families but expressed in different representational bases. This supports the view that the weak cross-family inheritance observed in Fig. 1 of main paper arises from representational misalignment rather than the absence of truthful heads. Moreover, it provides a mechanistic explanation: fine-tuning preserves functional subspaces within a family, while different pretraining lineages organize these subspaces differently, leading to distinct architectural patterns (e.g., head-level representations and their distributions) across families.
>
> > W2 & Q3 (the design of the soft gating mechanism)
>
> In our framework, we already employ layer- and head-specific probes, meaning that each attention head has its own learned Truth Score, and layer-specific truthful heads are implicitly identified during the probing stage. As a result, the gating mechanism operates on these fine-grained scores rather than treating all layers uniformly in terms of importance. While it is possible to introduce adaptive, learnable λ values per layer, our current design intentionally avoids additional training or tuning in the gating stage to keep the method simple and lightweight. Instead, we rely on the probe to capture layer-specific variations in truthfulness, and use a fixed λ to modulate contributions based on these scores.

---

### Official Review · Reviewer_WFS6 · 2026-03-12

**Soundness:** 3
**Presentation:** 3
**Significance:** 3
**Originality:** 3
**Overall Recommendation:** 4
**Confidence:** 2

**Summary:**

This paper investigates the inheritance of truthfulness-related attention heads from LLMs to their fine-tuned MLLMs. The authors propose TruthProbe, a method that:
+ Identifies "context-truthful heads" via linear probing on attention head outputs;
+ Demonstrates strong correlation (0.78-0.89) in truthfulness scores between base LLMs and their fine-tuned MLLM variants;
+ Introduces a soft-gating mechanism that modulates head contributions based on truthfulness scores.

**Compliance With Llm Reviewing Policy:**

Affirmed.

**Final Justification:**

My concerns have been addressed.

**Key Questions For Authors:**

See weakness.

**Limitations:**

yes

**Strengths And Weaknesses:**

Strengths:
+ The finding that truthfulness-related attention mechanisms are preserved across model families during multimodal fine-tuning is interesting.
+ The soft-gating mechanism is training-free during inference, and enables "plug-and-play" improvements across entire model families.

Weaknesses:
+ The paper does not provide an in-depth explanation of why truthful heads are preserved or what specific features these heads encode. A certain level of mechanistic interpretation would strengthen the contribution.
+ There may be differences in the definition of truthfulness between LLMs and MLLMs. Given the prevalent issue in MLLMs where visual information is abundant but effective information is scarce, further consideration should be given to attention heads that focus on irrelevant visual information.
+ The Truth Scores are derived from only 292 HaluEval samples (for LLMs) and 2,726 RLHF-V samples (for MLLMs). Although cross-validation was employed, statistical significance may be a concern. Experiments with larger and more diverse datasets are necessary.
+ The study only investigates two model families (Vicuna and Qwen2.5). It remains unclear whether these findings can be generalized to other architectures, such as models based on LLaMA-3, or closed-source models like Claude and GPT-style models.

---

> ### Author Rebuttal · Authors · 2026-03-31
>
> We thank the reviewer for the constructive feedback.
>
> > W1 (mechanistic interpretation)
>
> To better understand why truthful heads are preserved, we analyze the layer-wise weight differences between the base LLM, Vicuna-7B, and its multimodal variant, LLaVA-1.5. We measure layer-wise Frobenius norm of weight differences (Fig.1 in [Link](https://ibb.co/s9NRPqJS)), showing drift is concentrated in early layers.
>
> Prior work [1] suggests early layers process input signals, whereas deeper layers handle reasoning and high-level semantic integration. In our analysis, we find that truthful heads—identified by high Truth Scores—are predominantly located in middle to deeper layers (e.g., 80.0% of Top-20 truthful heads in LLaVA-1.5 are located in layers 10-31.), indicating that they are associated with context-level reasoning rather than low-level feature extraction. Combining these, we argue that truthful heads are preserved as they reside in minimally modified layers, leading to the observed inherited Truth Scores.
>
> We further quantify the overall representational similarity using the Frobenius norm of weight differences (Fig.2 in [Link](https://ibb.co/s9NRPqJS)). We find a clear contrast: within the same family (e.g., Vicuna-7B → LLaVA-1.5 / LLaVA-NeXT), the averaged Frobenius norm across all layers and heads is extremely small (≈0.03), indicating that head-level representations are largely preserved during fine-tuning. In contrast, across unrelated families (e.g., Vicuna vs. Mistral-7B), the norm is significantly larger (≈1.01), reflecting substantial representational divergence.
>
> This observation is also consistent with prior findings that Transformer fine-tuning tends to induce low-rank and localized updates, modifying only a small subset of parameters while preserving much of the original structure (e.g., [2] BitFit, [3] LoRA, [4] intrinsic dimensionality).
>
> [1] Attention Heads of Large Language Models (Zheng et al., 2024)
>
> [2] BitFit: Simple Parameter-efficient Fine-tuning for Transformer-based Masked Language-models (Zaken et al., 2021), ACL 2022
>
> [3] LoRA: Low-Rank Adaptation of Large Language Models (Hu et al., 2021)
>
> [4] Intrinsic Dimensionality of Common NLP Training Objectives (Aghajanyan et al., 2020), ACL 2021
>
> This result supports our interpretation from two perspectives: (1) within-family fine-tuning preserves head-level functions, explaining the inheritance of truthful heads, and (2) cross-family models exhibit fundamentally different representations, which is consistent with the low cross-family correlation of Truth Scores observed in our analysis.
>
>
> > W2 (the definition of truthfulness in MLLMs)
>
> While there may be apparent differences between LLMs and MLLMs, we argue that the definition of truthfulness is largely aligned: in both cases, models are given rich context (textual or visual), but effective information may be scarce for producing correct outputs. From this perspective, identifying heads that focus on relevant information is central to truthfulness in both settings.
> We agree that suppressing heads attending to irrelevant visual information is an interesting direction. In fact, our method implicitly captures this behavior.
> First, through probing, heads with high Truth Scores tend to focus on relevant information within the input.
> Second, our gating mechanism, based on these scores, naturally suppresses the contribution of low-scoring (non-truthful) heads, which can be interpreted as those attending to less relevant or misleading information.
> Thus, although not explicitly designed for visual relevance filtering, our framework already incorporates this effect in an intrinsic and indirect manner.
>
> > W3 (statistical significance)
> 1) HaluEval results as (Mean ± Std) for statistical stability
>
> We report the performance of HaluEval as the (Mean ± Std) in Tab.1 of [Link](https://ibb.co/s9NRPqJS), where three different random seeds are used for constructing HaluEval test set.
>
> 2) Robustness to Data Scale and Training Regimes for LLM/MLLM Probing
>
> To assess the robustness of our results to data scale and training configuration, we conduct additional experiments under substantially different probing regimes.
> For LLMs, we increase the dataset size from 292 samples × 200 epochs to 2,920 samples × 20 epochs, and evaluate on the remaining 7,080 samples from HaluEval. For MLLMs, we reduce the dataset size from 2,726 samples × 200 epochs to 273 samples × 2,000 epochs and evaluate on POPE (COCO) and CHAIR benchmarks.
> As shown in Tab.2~4 (in [Link](https://ibb.co/s9NRPqJS)), the performance gains remain consistent with those reported in the main submission, demonstrating that our method is robust to variations in data scale and training regimes, and alleviating concerns about overfitting or instability due to limited sample sizes.
>
> > W4 (model coverage)
>
> We also verified the effects of our method on additional model families; due to space limitations, please refer to our response to Reviewer 4Zjf.

---

> > ### Author Rebuttal · Reviewer_WFS6 · 2026-04-02
> >
> > Thanks for the rebuttal, which has addressed many of my concerns. I have two follow-up questions:
> > 1. For W1,  the rebuttal focuses on where truthful heads are located (layer depth) and how much weight change occurs, but it does not answer what these heads actually do mechanistically. I am curious whether they are really capturing some features like “confidence” or “source credibility”, or whether their attention patterns differ from non‑truthful heads?
> > 2. For W3, I meant to suggest that a significant test should be conducted.

---

> > > ### Author Response · Authors · 2026-04-03
> > >
> > > > For W1, the rebuttal focuses on where truthful heads are located (layer depth) and how much weight change occurs, but it does not answer what these heads actually do mechanistically. I am curious whether they are really capturing some features like “confidence” or “source credibility”, or whether their attention patterns differ from non‑truthful heads?
> > >
> > > We thank the reviewer for this insightful question. We agree that understanding whether truthful heads capture meaningful signals such as confidence or source credibility is crucial for a mechanistic interpretation.
> > >
> > > To investigate this, we analyze **where different heads attend in the image** by visualizing their attention patterns. A direct visualization of the original attention maps, however, is dominated by *attention sink tokens*, i.e., tokens that consistently receive high attention regardless of the query. This phenomenon has been discussed in prior work [1], which shows that Transformers tend to aggregate attention into a small set of irrelevant tokens. As a result, raw attention maps obscure head-specific behaviors.
> > >
> > > To address this issue, we adopt the *relative attention* formulation proposed in [2]. Specifically, we normalize the attention induced by a given query (e.g., *"Who is wearing the dress?"*) with respect to a general query (e.g., *"Write a general description of the image."*), allowing us to isolate query-dependent attention patterns.
> > >
> > > We visualize both attention maps (obtained from LLaVA-1.5) and their overlays on the image (please refer this [**Link**](https://ibb.co/3mxZzDWD)). Using the Truth Score (obtained by probing the base LLM, Vicuna-7B), we compare Top-k (k=3) truthful heads and Bottom-k (k=3) non-truthful heads in LLaVA-1.5. The visualization corresponds to attention from the final query token to visual tokens (24×24 grid).
> > >
> > > From these results, we observe the following:
> > >
> > > 1. **Truthful heads exhibit semantically meaningful and query-dependent attention.**
> > >
> > > They focus on regions directly relevant to the query (e.g., the referenced object or person) and show spatially selective patterns that depend on the query. This indicates their role in grounding the model’s responses by attending to *credible visual evidence*, rather than amplifying generic features.
> > >
> > > 2. **Non-truthful heads exhibit non-semantic, position-dependent patterns.**
> > >
> > > Their attention forms diagonal/striped structures that are largely invariant to the query. As these heads are primarily in early layers(e.g., layer 0), where cross-modal alignment is weak, their behavior reflects positional biases (likely driven by positional encoding) in the visual grid rather than meaningful visual grounding.
> > >
> > > Overall, this comparison highlights a clear mechanistic distinction: **truthful heads demonstrate query-dependent, evidence-focused attention**, whereas **non-truthful heads largely reflect early-stage representation processing** rather than direct semantic grounding.
> > >
> > > [1] Vision Transformers Need Registers (Timothée et al., ICLR 2024)
> > >
> > > [2] MLLMs Know Where to Look: Training-free Perception of Small Visual Details with Multimodal LLMs (Jiarui et al., ICLR 2025)
> > >
> > > > For W3, I meant to suggest that a significant test should be conducted.
> > >
> > > We thank the reviewer for highlighting the importance of statistical validation. Following this suggestion, we conducted a **paired t-test** to verify **whether the observed performance improvements are statistically significant**.
> > >
> > > Specifically, we evaluate both a base LLM (Qwen2.5-7B), and its corresponding MLLMs (Qwen2.5-VL-Instruct and Qwen2.5-VL-Omni). We use benchmarks that provide per-sample accuracy signals, namely HaluEval and POPE (COCO). For each setting, we compute per-sample accuracy differences between the baseline and our method, and perform a paired t-test to assess whether the mean improvement is significantly greater than zero.
> > >
> > > Across all settings, we observe statistically significant improvements. In particular, for Qwen2.5-7B on HaluEval, the mean per-sample difference is 0.0778 with an extremely small p-value (p ≈ 1.45e-67), indicating a highly robust and consistent gain. For the MLLMs (Qwen2.5-VL-Instruct and Qwen2.5-VL-Omni), although the average per-sample differences are relatively small (+0.0049 and +0.0199), they remain statistically significant, with p = 0.0004 and p = 3.04e-26, respectively. This indicates that the improvements are consistently observed across samples rather than arising from random fluctuations.
> > >
> > > These results confirm that the **observed performance gains are not due to chance, but reflect statistically reliable improvements across both LLM and MLLM settings**.

---

### Official Review · Reviewer_HJSY · 2026-03-12

**Soundness:** 2
**Presentation:** 3
**Significance:** 3
**Originality:** 2
**Overall Recommendation:** 4
**Confidence:** 3

**Summary:**

This paper studies whether truthfulness is inherited within a model family when a base LLM is adapted into multimodal models. It identifies attention heads related to context-faithful generation, shows that these head-level truthfulness scores are similar within the same family, and proposes a soft-gating method that uses these scores to improve truthfulness and reduce hallucination in descendant models.

**Compliance With Llm Reviewing Policy:**

Affirmed.

**Final Justification:**

My concerns have been addressed.

**Key Questions For Authors:**

1. Can you quantify the practical benefit of family-level transfer over directly probing each target MLLM (e.g., compute, data, and tuning cost)? A clear cost/sample-efficiency advantage would make the contribution much stronger in practice.
2. Does the proposed mechanism help on more complex multimodal reasoning tasks beyond the current hallucination-focused benchmarks? Positive evidence on more reasoning-intensive settings would make me view the contribution as broader than benchmark-specific hallucination mitigation.
3. How robust is the inheritance claim across more model scales or additional families? Additional validation beyond the current scope would make the central claim feel much more general and impactful.

**Limitations:**

yes

**Strengths And Weaknesses:**

# Strength
1. The paper studies an important question: whether multimodal models keep some basic behavior from their base LLMs. This is useful and relevant, especially for context-faithfulness and hallucination reduction
2. The main finding is interesting: some attention heads seem related to context-faithful behavior, and this pattern stays similar within the same model family.
3. Based on this finding, the paper proposes a soft-gating method that uses truthfulness scores from the base LLM to guide later models. This gives extra support to the paper’s main idea.
4. The paper tests its idea in both single-dataset and cross-dataset settings, which makes the evidence more convincing

# Weaknesses
1. Unclear Practical Benefit: The paper does not clearly show the real practical value of this finding. In many results, using scores from the base LLM is only close to directly probing the target MLLM, not clearly better. therefore it is still unclear what the real gain is in practice. The authors should give more numbers on cost, data efficiency, and stability.
2. Limited Analysis of Unexpected Results: Some results are not explained well enough. In several cases, using the base LLM’s scores works better than using the target MLLM’s own scores, which is a bit surprising. The paper should discuss this more carefully, for example with case studies or more analysis.
3. Limited Model Coverage: The experiments are still limited in model type and scale. Most results are based on a small set of related model families, mainly around the same size. It is still unclear whether the same finding holds for more model types or much larger/smaller models.
4. Limited Task Range: The task range is still somewhat limited. The paper includes QA-style tasks and image description, but it is still unclear whether the method would help on harder multimodal reasoning tasks.
5. The paper would be stronger if it tested more model families. That would help show whether this inheritance effect is general, rather than only true for the families used in the paper.

---

> ### Author Rebuttal · Authors · 2026-03-31
>
> We thank the reviewer for providing thorough feedback regarding the practical value, model / task coverage, and the analysis of results.
>
> > W1. Unclear Practical Benefit: The paper does not clearly show the real practical value of this finding. In many results, using scores from the base LLM is only close to directly probing the target MLLM, not clearly better. therefore it is still unclear what the real gain is in practice. The authors should give more numbers on cost, data efficiency, and stability.
>
> > Q1. Can you quantify the practical benefit of family-level transfer over directly probing each target MLLM (e.g., compute, data, and tuning cost)? A clear cost/sample-efficiency advantage would make the contribution much stronger in practice.
>
> While directly probing a target MLLM is a straightforward approach and can achieve slightly better performance, our method serves as an effective proxy by providing comparable results while offering practical advantages from two perspectives.
>
> First, probing MLLMs incurs substantially higher end-to-end cost compared to LLM probing in the overall probing pipeline, including activation extraction and prober training. For 10,000 samples (text-only for LLMs vs. image-text pairs for MLLMs), we observe that LLaVA-1.5 and LLaVA-NeXT require approximately **5.6× and 21.8× more TFLOPs**, respectively, than the base LLM Vicuna-7B.
>
> Second, and more importantly, our method enables a **“probe-once, reuse-within-family”** paradigm, which eliminates the need to repeat the full probing pipeline—including data curation, activation extraction, and probe training/validation—for each new model variant. As modern MLLMs are frequently released in multiple fine-tuned versions, this avoids redundant computation across versions and leads to a system-level reduction in total computational cost.
>
> In summary, the practical benefit of our approach lies in achieving comparable performance while substantially reducing the cumulative cost of repeated probing pipelines across model variants.
>
> > W2. Limited Analysis of Unexpected Results: Some results are not explained well enough. In several cases, using the base LLM’s scores works better than using the target MLLM’s own scores, which is a bit surprising. The paper should discuss this more carefully, for example with case studies or more analysis.
>
> One possible explanation is that probing an MLLM is inherently more challenging than probing an LLM due to the multimodal nature of its inputs. In LLM probing, the model only needs to focus on textual context, allowing the probe to more cleanly capture signals related to contextual truthfulness. In contrast, MLLMs must process both visual and textual information, where a large portion of visual input may be irrelevant or noisy for the task at hand. As a result, the learned probe for MLLMs may be less precise in isolating the features that correspond to truthfulness, since the representation space mixes relevant and irrelevant signals across modalities. This can lead to cases where Truth Scores derived from a base LLM provide a more reliable estimate of contextual truthfulness than those obtained directly from the target MLLM. We will further clarify this point in the revision and provide additional analysis to better understand this behavior.
>
> **More model families and sizes (W3, W5, Q3) and more task range (W4, Q2)**
> Due to the response length limit, we provide detailed results of this feedback in our response to Reviewer 4Zjf. Please refer to that section for additional results and discussion.

---

> > ### Author Rebuttal · Reviewer_HJSY · 2026-04-01
> >
> > Thank you for your thoughtful rebuttal; I understood the key points. I believe this paper offers strong practical value for understanding the relationship between MLLMs and their base LLMs, and incorporating these additional results makes the argument significantly more comprehensive and well-rounded.  I will be raising my score to 4 (weak accept).

---

> > > ### Author Response · Authors · 2026-04-02
> > >
> > > We sincerely thank the reviewer for the constructive feedback and the time devoted to evaluating our manuscript and rebuttal. Your suggestions have been highly valuable in improving the depth and overall quality of our work.
> > >
> > > We also sincerely appreciate your positive reassessment of our paper.
> > > All additional clarifications and experiments will be incorporated into the final version.

---

### Decision · Program_Chairs · 2026-04-30

**Decision:**

Accept (regular)

**Comment:**

This paper proposes TruthProbe, a linear probing method that quantifies "Truth Scores" at the attention head level, revealing strong correlations in truthfulness scores between base LLMs and their multimodal descendants within the same model family. Based on this phenomenon, they introduce a soft-gating mechanism that amplifies the influence of context-truthful heads during inference.  Experiments on HaluEval, POPE, and CHAIR benchmarks demonstrate that Truth Scores derived from a base LLM can be effectively transferred to MLLMs as a plug-and-play gate, achieving performance gains comparable to probing the target MLLMs directly.

The reviewers identify several notable strengths. Examining model reliability from a lineage-level perspective is interesting and underexplored. The empirical finding of truthfulness could provide practical value. The proposed mechanism is lightweight and training-free during inference. However, reviewers raised several major concerns. Reviewer HJsy questioned the practical benefit and requested quantification of computational cost advantages. Analysis of unexpected results where base LLM scores outperform MLLM scores is limited. Reviewer WFS6 highlighted the lack of mechanistic interpretation. Reviewer ym5Q raised concerns about generalization to low-resource domains, hyperparameter selection, and insufficient empirical validation. Reviewer 4Zif noted that the evaluated models are relatively small in scale and outdated.

During the rebuttal phase, the authors provided additional evidence addressing most concerns. Regarding practical benefits, they quantified computational costs. For mechanistic interpretation, they provided layer-wise weight difference analysis showing that drift is concentrated in early layers while truthful heads reside in minimally modified middle-to-deep layers. On model coverage, they extended experiments to Qwen3-8B/VL and InternLM3-8B/InternVL3-9B families with consistent improvements. For task range, they added results on GQA and MMMU benchmarks. Regarding low-resource domains, they demonstrated successful transfer to a medical domain. Most reviewers are satisfied with the rebuttal. Reviewer ym5Q acknowledged the additional experiments but maintained that empirical validation remained insufficient.

Overall, this paper presents technically sound work with an interesting and underexplored perspective on model lineage inheritance. The authors have addressed the majority of concerns through extensive supplementary experiments and analysis during the rebuttal period.